# ALGORITHMIC FRAMEWORK FOR MODEL-BASED DEEP REINFORCEMENT LEARNING WITH THEORETICAL GUARANTEES

Yuping Luo [*1], Huazhe Xu [*2], Yuanzhi Li[4], Yuandong Tian[3], Trevor Darrell[2], and Tengyu Ma[4]

[1]Princeton University, `yupingl@cs.princeton.edu`
[2]University of California, Berkeley, `{huazhe_xu,trevor}@eecs.berkeley.edu`
[3]Facebook AI Research, `yuandong@fb.com`
[4]Stanford University. `{yuanzhil,tengyuma}@stanford.edu`

## ABSTRACT

Model-based reinforcement learning (RL) is considered to be a promising approach to reduce the sample complexity that hinders model-free RL. However, the theoretical understanding of such methods has been rather limited. This paper introduces a novel algorithmic framework for designing and analyzing model-based RL algorithms with theoretical guarantees. We design a meta-algorithm with a theoretical guarantee of monotone improvement to a local maximum of the expected reward. The meta-algorithm iteratively builds a lower bound of the expected reward based on the estimated dynamical model and sample trajectories, and then maximizes the lower bound jointly over the policy and the model. The framework extends the optimism-in-face-of-uncertainty principle to non-linear dynamical models in a way that requires *no explicit* uncertainty quantification. Instantiating our framework with simplification gives a variant of model-based RL algorithms Stochastic Lower Bounds Optimization (SLBO). Experiments demonstrate that SLBO achieves state-of-the-art performance when only one million or fewer samples are permitted on a range of continuous control benchmark tasks.[1]

## 1 INTRODUCTION

In recent years deep reinforcement learning has achieved strong empirical success, including super-human performances on Atari games and Go (Mnih et al., 2015; Silver et al., 2017) and learning locomotion and manipulation skills in robotics (Levine et al., 2016; Schulman et al., 2015b; Lillicrap et al., 2015). Many of these results are achieved by model-free RL algorithms that often require a massive number of samples, and therefore their applications are mostly limited to simulated environments. Model-based deep reinforcement learning, in contrast, exploits the information from state observations explicitly — by planning with an estimated dynamical model — and is considered to be a promising approach to reduce the sample complexity. Indeed, empirical results (Deisenroth & Rasmussen, 2011b; Deisenroth et al., 2013; Levine et al., 2016; Nagabandi et al., 2017; Kurutach et al., 2018; Pong et al., 2018a) have shown strong improvements in sample efficiency.

Despite promising empirical findings, many of *theoretical* properties of model-based deep reinforcement learning are not well-understood. For example, how does the error of the estimated model affect the estimation of the value function and the planning? Can model-based RL algorithms be guaranteed to improve the policy monotonically and converge to a local maximum of the value function? How do we quantify the uncertainty in the dynamical models?

It's challenging to address these questions theoretically in the context of deep RL with continuous state and action space and non-linear dynamical models. Due to the high-dimensionality, learning models from observations in one part of the state space and extrapolating to another part sometimes

---

[0]* indicates equal contribution
[1]The source code of this work is available at https://github.com/roosephu/slbo

involves a leap of faith. The uncertainty quantification of the non-linear parameterized dynamical models is difficult — even without the RL components, it is an active but widely-open research area. Prior work in model-based RL mostly quantifies uncertainty with either heuristics or simpler models (Moldovan et al., 2015; Xie et al., 2016; Deisenroth & Rasmussen, 2011a).

Previous theoretical work on model-based RL mostly focuses on either the finite-state MDPs (Jaksch et al., 2010; Bartlett & Tewari, 2009; Fruit et al., 2018; Lakshmanan et al., 2015; Hinderer, 2005; Pirotta et al., 2015; 2013), or the linear parametrization of the dynamics, policy, or value function (Abbasi-Yadkori & Szepesvári, 2011; Simchowitz et al., 2018; Dean et al., 2017; Sutton et al., 2012; Tamar et al., 2012), but not much on non-linear models. Even with an oracle prediction intervals[2] or posterior estimation, to the best of our knowledge, there was no previous algorithm with convergence guarantees for model-based deep RL.

Towards addressing these challenges, the main contribution of this paper is to propose a novel algorithmic framework for model-based deep RL with theoretical guarantees. Our meta-algorithm (Algorithm 1) extends the optimism-in-face-of-uncertainty principle to non-linear dynamical models in a way that requires *no explicit* uncertainty quantification of the dynamical models.

Let $V^\pi$ be the value function $V^\pi$ of a policy $\pi$ on the true environment, and let $\widehat{V}^\pi$ be the value function of the policy $\pi$ on the estimated model $\widehat{M}$. We design provable upper bounds, denoted by $D^{\pi,\widehat{M}}$, on how much the error can compound and divert the expected value $\widehat{V}^\pi$ of the imaginary rollouts from their real value $V^\pi$, in a neighborhood of some reference policy. Such upper bounds capture the intrinsic difference between the estimated and real dynamical model with respect to the particular reward function under consideration.

The discrepancy bounds $D^{\pi,\widehat{M}}$ naturally leads to a lower bound for the true value function:

$$V^\pi \geq \widehat{V}^\pi - D^{\pi,\widehat{M}}. \tag{1.1}$$

Our algorithm iteratively collects batches of samples from the interactions with environments, builds the lower bound above, and then maximizes it over both the dynamical model $\widehat{M}$ and the policy $\pi$. We can use any RL algorithms to optimize the lower bounds, because it will be designed to only depend on the sample trajectories from a fixed reference policy (as opposed to requiring new interactions with the policy iterate.)

We show that the performance of the policy is guaranteed to monotonically increase, assuming the optimization within each iteration succeeds (see Theorem 3.1.) To the best of our knowledge, this is the first theoretical guarantee of monotone improvement for model-based deep RL.

Readers may have realized that optimizing a robust lower bound is reminiscent of robust control and robust optimization. The distinction is that we optimistically and iteratively maximize the RHS of (1.1) jointly over the model and the policy. The iterative approach allows the algorithms to collect higher quality trajectory adaptively, and the optimism in model optimization encourages explorations of the parts of space that are not covered by the current discrepancy bounds.

To instantiate the meta-algorithm, we design a few valid discrepancy bounds in Section 4. In Section 4.1, we recover the norm-based model loss by imposing the additional assumption of a Lipschitz value function. The result suggests a norm is preferred compared to the square of the norm. Indeed in Section 6.2, we show that experimentally learning with $\ell_2$ loss significantly outperforms the mean-squared error loss ($\ell_2^2$).

In Section 4.2, we design a discrepancy bound that is *invariant* to the representation of the state space. Here we measure the loss of the model by the difference between the value of the predicted next state and the value of the true next state. Such a loss function is shown to be invariant to one-to-one transformation of the state space. Thus we argue that the loss is an intrinsic measure for the model error without any information beyond observing the rewards. We also refine our bounds in Section A by utilizing some mathematical tools of measuring the difference between policies in $\chi^2$-divergence (instead of KL divergence or TV distance).

---

[2]We note that the confidence interval of parameters are likely meaningless for over-parameterized neural networks models.

Our analysis also sheds light on the comparison between model-based RL and on-policy model-free RL algorithms such as policy gradient or TRPO (Schulman et al., 2015a). The RHS of equation (1.1) is likely to be a good approximator of $V^\pi$ in a larger neighborhood than the linear approximation of $V^\pi$ used in policy gradient is (see Remark 4.5.)

Finally, inspired by our framework and analysis, we design a variant of model-based RL algorithms Stochastic Lower Bounds Optimization (SLBO). Experiments demonstrate that SLBO achieves state-of-the-art performance when only 1M samples are permitted on a range of continuous control benchmark tasks.

## 2   NOTATIONS AND PRELIMINARIES

We denote the state space by $\mathcal{S}$, the action space by $\mathcal{A}$. A policy $\pi(\cdot|s)$ specifies the conditional distribution over the action space given a state $s$. A dynamical model $M(\cdot|s, a)$ specifies the conditional distribution of the next state given the current state $s$ and action $a$. We will use $M^\star$ globally to denote the unknown true dynamical model. Our target applications are problems with the continuous state and action space, although the results apply to discrete state or action space as well. When the model is deterministic, $M(\cdot|s, a)$ is a dirac measure. In this case, we use $M(s, a)$ to denote the unique value of $s'$ and view $M$ as a function from $\mathcal{S} \times \mathcal{A}$ to $\mathcal{S}$. Let $\mathcal{M}$ denote a (parameterized) family of models that we are interested in, and $\Pi$ denote a (parameterized) family of policies.

Unless otherwise stated, for random variable $X$, we will use $p_X$ to denote its density function.

Let $S_0$ be the random variable for the initial state. Let $S_t^{\pi,M}$ to denote the random variable of the states at steps $t$ when we execute the policy $\pi$ on the dynamic model $M$ stating with $S_0$. Note that $S_0^{\pi,M} = S_0$ unless otherwise stated. We will omit the subscript when it's clear from the context. We use $A_t$ to denote the actions at step $t$ similarly. We often use $\tau$ to denote the random variable for the trajectory $(S_0, A_1, \ldots, S_t, A_t, \ldots)$. Let $R(s, a)$ be the reward function at each step. We assume $R$ is *known* throughout the paper, although $R$ can be also considered as part of the model if unknown. Let $\gamma$ be the discount factor.

Let $V^{\pi,M}$ be the value function on the model $M$ and policy $\pi$ defined as:

$$V^{\pi,M}(s) = \mathbb{E}_{\substack{\forall t \geq 0, A_t \sim \pi(\cdot|S_t) \\ S_{t+1} \sim M(\cdot|S_t, A_t)}} \left[ \sum_{t=0}^{\infty} \gamma^t R(S_t, A_t) \mid S_0 = s \right] \tag{2.1}$$

We define $V^{\pi,M} = \mathbb{E}\left[ V^{\pi,M}(S_0) \right]$ as the expected reward-to-go at Step 0 (averaged over the random initial states). Our goal is to maximize the reward-to-go on the true dynamical model, that is, $V^{\pi,M^\star}$, over the policy $\pi$. For simplicity, throughout the paper, we set $\kappa = \gamma(1 - \gamma)^{-1}$ since it occurs frequently in our equations. Every policy $\pi$ induces a distribution of states visited by policy $\pi$:

**Definition 2.1.** For a policy $\pi$, define $\rho^{\pi,M}$ as the discounted distribution of the states visited by $\pi$ on $M$. Let $\rho^\pi$ be a shorthand for $\rho^{\pi,M^\star}$ and we omit the superscript $M^\star$ throughout the paper. Concretely, we have $\rho^\pi = (1 - \gamma) \sum_{t=0}^{\infty} \gamma^t \cdot p_{S_t^\pi}$

## 3   ALGORITHMIC FRAMEWORK

As mentioned in the introduction, towards optimizing $V^{\pi,M^\star}$,[3] our plan is to build a lower bound for $V^{\pi,M^\star}$ of the following type and optimize it iteratively:

$$V^{\pi,M^\star} \geq V^{\pi,\widehat{M}} - D(\widehat{M}, \pi) \tag{3.1}$$

where $D(\widehat{M}, \pi) \in \mathbb{R}_{\geq 0}$ bounds from above the discrepancy between $V^{\pi,\widehat{M}}$ and $V^{\pi,M^\star}$. Building such an optimizable discrepancy bound globally that holds for all $\widehat{M}$ and $\pi$ turns out to be rather difficult, if not impossible. Instead, we shoot for establishing such a bound over the neighborhood of a reference policy $\pi_{\text{ref}}$.

$$V^{\pi,M^\star} \geq V^{\pi,\widehat{M}} - D_{\pi_{\text{ref}},\delta}(\widehat{M}, \pi), \qquad \forall \pi \text{ s.t. } d(\pi, \pi_{\text{ref}}) \leq \delta \tag{R1}$$

---

[3]Note that in the introduction we used $V^\pi$ for simplicity, and in the rest of the paper we will make the dependency on $M^\star$ explicit.

Here $d(\cdot, \cdot)$ is a function that measures the closeness of two policies, which will be chosen later in alignment with the choice of $D$. We will mostly omit the subscript $\delta$ in $D$ for simplicity in the rest of the paper. We will require our discrepancy bound to vanish when $\widehat{M}$ is an accurate model:

$$\widehat{M} = M^\star \Longrightarrow D_{\pi_{\text{ref}}}(\widehat{M}, \pi) = 0, \quad \forall \pi, \pi_{\text{ref}} \tag{R2}$$

The third requirement for the discrepancy bound $D$ is that it can be estimated and optimized in the sense that

$$D_{\pi_{\text{ref}}}(\widehat{M}, \pi) \text{ is of the form } \underset{\tau \sim \pi_{\text{ref}}, M^\star}{\mathbb{E}} [f(\widehat{M}, \pi, \tau)] \tag{R3}$$

where $f$ is a known differentiable function. We can estimate such discrepancy bounds for *every* $\pi$ in the neighborhood of $\pi_{\text{ref}}$ by sampling empirical trajectories $\tau^{(1)}, \ldots, \tau^{(n)}$ from executing policy $\pi_{\text{ref}}$ on the real environment $M^\star$ and compute the average of $f(\widehat{M}, \pi, \tau^{(i)})$'s. We would have to insist that the expectation cannot be over the randomness of trajectories from $\pi$ on $M^\star$, because then we would have to re-sample trajectories for every possible $\pi$ encountered.

For example, assuming the dynamical models are all deterministic, one of the valid discrepancy bounds (under some strong assumptions) that will prove in Section 4 is a multiple of the error of the prediction of $\widehat{M}$ on the trajectories from $\pi_{\text{ref}}$:

$$D_{\pi_{\text{ref}}}(\widehat{M}, \pi) = L \cdot \underset{S_0, \ldots, S_t, \sim \pi_{\text{ref}}, M^\star}{\mathbb{E}} \left[ \|\widehat{M}(S_t) - S_{t+1}\| \right] \tag{3.2}$$

Suppose we can establish such an discrepancy bound $D$ (and the distance function $d$) with properties (R1), (R2), and (R3), — which will be the main focus of Section 4 —, then we can devise the following meta-algorithm (Algorithm 1). We iteratively optimize the lower bound over the policy $\pi_{k+1}$ and the model $M_{k+1}$, subject to the constraint that the policy is not very far from the reference policy $\pi_k$ obtained in the previous iteration. For simplicity, we only state the population version with the exact computation of $D_{\pi_{\text{ref}}}(\widehat{M}, \pi)$, though empirically it is estimated by sampling trajectories.

---

**Algorithm 1** Meta-Algorithm for Model-based RL

---

**Inputs:** Initial policy $\pi_0$. Discrepancy bound $D$ and distance function $d$ that satisfy equation (R1) and (R2).
**For** $k = 0$ to $T$:

$$\pi_{k+1}, M_{k+1} = \underset{\pi \in \Pi, M \in \mathcal{M}}{\text{argmax}} \quad V^{\pi, M} - D_{\pi_k, \delta}(M, \pi) \tag{3.3}$$

$$\text{s.t. } d(\pi, \pi_k) \leq \delta \tag{3.4}$$

---

We first remark that the discrepancy bound $D_{\pi_k}(M, \pi)$ in the objective plays the role of learning the dynamical model by ensuring the model to fit to the sampled trajectories. For example, using the discrepancy bound in the form of equation (3.2), we roughly recover the standard objective for model learning, with the caveat that we only have the norm instead of the square of the norm in MSE. Such distinction turns out to be empirically important for better performance (see Section 6.2).

Second, our algorithm can be viewed as an extension of the optimism-in-face-of-uncertainty (OFU) principle to non-linear parameterized setting: jointly optimizing $M$ and $\pi$ encourages the algorithm to choose the most optimistic model among those that can be used to accurately estimate the value function. (See (Jaksch et al., 2010; Bartlett & Tewari, 2009; Fruit et al., 2018; Lakshmanan et al., 2015; Pirotta et al., 2015; 2013) and references therein for the OFU principle in finite-state MDPs.) The main novelty here is to optimize the lower bound directly, without explicitly building any confidence intervals, which turns out to be challenging in deep learning. In other words, the uncertainty is measured straightforwardly by how the error would affect the estimation of the value function.

Thirdly, the maximization of $V^{\pi, M}$, when $M$ is fixed, can be solved by any model-free RL algorithms with $M$ as the environment without querying any real samples. Optimizing $V^{\pi, M}$ jointly over $\pi, M$ can be also viewed as another RL problem with an extended actions space using the known "extended MDP technique". See (Jaksch et al., 2010, section 3.1) for details.

Our main theorem shows formally that the policy performance in the real environment is non-decreasing under the assumption that the real dynamics belongs to our parameterized family $\mathcal{M}$.[4]

**Theorem 3.1.** *Suppose that $M^\star \in \mathcal{M}$, that $D$ and $d$ satisfy equation* (R1) *and* (R2)*, and the optimization problem in equation* (3.3) *is solvable at each iteration. Then, Algorithm 1 produces a sequence of policies $\pi_0, \ldots, \pi_T$ with monotonically increasing values:*

$$V^{\pi_0, M^\star} \leq V^{\pi_1, M^\star} \leq \cdots \leq V^{\pi_T, M^\star} \tag{3.5}$$

*Moreover, as $k \to \infty$, the value $V^{\pi_k, M^\star}$ converges to some $V^{\bar{\pi}, M^\star}$, where $\bar{\pi}$ is a local maximum of $V^{\pi, M^\star}$ in domain $\Pi$.*

The theorem above can also be extended to a finite sample complexity result with standard concentration inequalities. We show in Theorem G.2 that we can obtain an approximate local maximum in $O(1/\varepsilon)$ iterations with sample complexity (in the number of trajectories) that is polynomial in dimension and accuracy $\varepsilon$ and is logarithmic in certain smoothness parameters.

*Proof of Theorem 3.1.* Since $D$ and $d$ satisfy equation (R1), we have that

$$V^{\pi_{k+1}, M^\star} \geq V^{\pi_{k+1}, M_{k+1}} - D_{\pi_k}(M_{k+1}, \pi_{k+1})$$

By the definition that $\pi_{k+1}$ and $M_{k+1}$ are the optimizers of equation (3.3), we have that

$$V^{\pi_{k+1}, M_{k+1}} - D_{\pi_k}(M_{k+1}, \pi_{k+1}) \geq V^{\pi_k, M^\star} - D_{\pi_k}(M^\star, \pi_k) = V^{\pi_k, M^\star} \quad \text{(by equation R2)}$$

Combing the two equations above we complete the proof of equation (3.5).

For the second part of the theorem, by compactness, we have that a subsequence of $\pi_k$ converges to some $\bar{\pi}$. By the monotonicity we have $V^{\pi_k, M^\star} \leq V^{\bar{\pi}, M^\star}$ for every $k \geq 0$. For the sake of contradiction, we assume $\bar{\pi}$ is a not a local maximum, then in the neighborhood of $\bar{\pi}$ there exists $\pi'$ such that $V^{\pi', M^\star} > V^{\bar{\pi}, M^\star}$ and $d(\bar{\pi}, \pi') < \delta/2$. Let $t$ be such that $\pi_t$ is in the $\delta/2$-neighborhood of $\bar{\pi}$. Then we see that $(\pi', M^\star)$ is a better solution than $(\pi_{t+1}, M_{t+1})$ for the optimization problem (3.3) in iteration $t$ because $V^{\pi', M^\star} > V^{\bar{\pi}, M^\star} \geq V^{\pi_{t+1}, M^\star} \geq V^{\pi_{t+1}, M_{t+1}} - D_{\pi_t}(M_{t+1}, \pi_{t+1})$. (Here the last inequality uses equation (R1) with $\pi_t$ as $\pi_{\text{ref}}$.) The fact $(\pi', M^\star)$ is a strictly better solution than $(\pi_{t+1}, M_{t+1})$ contradicts the fact that $(\pi_{t+1}, M_{t+1})$ is defined to be the optimal solution of (3.3) . Therefore $\bar{\pi}$ is a local maximum and we complete the proof.

$\square$

## 4 Discrepancy Bounds Design

In this section, we design discrepancy bounds that can provably satisfy the requirements (R1), (R2), and (R3). We design increasingly stronger discrepancy bounds from Section 4.1 to Section A.

### 4.1 Norm-based prediction error bounds

In this subsection, we assume the dynamical model $M^\star$ is deterministic and we also learn with a deterministic model $\widehat{M}$. Under assumptions defined below, we derive a discrepancy bound $D$ of the form $\|\widehat{M}(S, A) - M^\star(S, A)\|$ averaged over the observed state-action pair $(S, A)$ on the dynamical model $\widehat{M}$. This suggests that the norm is a better metric than the mean-squared error for learning the model, which is empirically shown in Section 6.2. Through the derivation, we will also introduce a telescoping lemma, which serves as the main building block towards other finer discrepancy bounds.

We make the (strong) assumption that the value function $V^{\pi, \widehat{M}}$ on the estimated dynamical model is $L$-Lipschitz w.r.t to some norm $\|\cdot\|$ in the sense that

$$\forall s, s' \in \mathcal{S}, \left|V^{\pi, \widehat{M}}(s) - V^{\pi, \widehat{M}}(s')\right| \leq L \cdot \|s - s'\| \tag{4.1}$$

In other words, nearby starting points should give reward-to-go under the same policy $\pi$. We note that not every real environment $M^\star$ has this property, let alone the estimated dynamical models.

---

[4]We note that such an assumption, though restricted, may not be very far from reality: optimistically speaking, we only need to approximate the dynamical model accurately on the trajectories of the optimal policy. This might be much easier than approximating the dynamical model globally.

However, once the real dynamical model induces a Lipschitz value function, we may penalize the Lipschitz-ness of the value function of the estimated model during the training.

We start off with a lemma showing that the expected prediction error is an upper bound of the discrepancy between the real and imaginary values.

**Lemma 4.1.** *Suppose $V^{\pi,\widehat{M}}$ is L-Lipschitz (in the sense of (4.1)). Recall $\kappa = \gamma(1-\gamma)^{-1}$.*

$$\left| V^{\pi,\widehat{M}} - V^{\pi,M^\star} \right| \le \kappa L \underset{\substack{S\sim\rho^\pi \\ A\sim\pi(\cdot|S)}}{\mathbb{E}} \left[ \|\widehat{M}(S,A) - M^\star(S,A)\| \right] \tag{4.2}$$

However, in RHS in equation 4.2 cannot serve as a discrepancy bound because it does not satisfy the requirement (R3) — to optimize it over $\pi$ we need to collect samples from $\rho^\pi$ for every iterate $\pi$ — the state distribution of the policy $\pi$ on the *real* model $M^\star$. The main proposition of this subsection stated next shows that for every $\pi$ in the neighborhood of a reference policy $\pi_{\text{ref}}$, we can replace the distribution $\rho^\pi$ be a fixed distribution $\rho^{\pi_{\text{ref}}}$ with incurring only a higher order approximation. We use the expected KL divergence between two $\pi$ and $\pi_{\text{ref}}$ to define the neighborhood:

$$d^{\text{KL}}(\pi, \pi_{\text{ref}}) = \underset{S\sim\rho^\pi}{\mathbb{E}} \left[ KL(\pi(\cdot|S), \pi_{\text{ref}}(\cdot|S))^{1/2} \right] \tag{4.3}$$

**Proposition 4.2.** *In the same setting of Lemma 4.1, assume in addition that $\pi$ is close to a reference policy $\pi_{\text{ref}}$ in the sense that $d^{\text{KL}}(\pi, \pi_{\text{ref}}) \le \delta$, and that the states in $\mathcal{S}$ are uniformly bounded in the sense that $\|s\| \le B, \forall s \in \mathcal{S}$. Then,*

$$\left| V^{\pi,\widehat{M}} - V^{\pi,M^\star} \right| \le \kappa L \underset{\substack{S\sim\rho^{\pi_{\text{ref}}} \\ A\sim\pi(\cdot|S)}}{\mathbb{E}} \left[ \|\widehat{M}(S,A) - M^\star(S,A)\| \right] + 2\kappa^2\delta B \tag{4.4}$$

In a benign scenario, the second term in the RHS of equation (4.4) should be dominated by the first term when the neighborhood size $\delta$ is sufficiently small. Moreover, the term $B$ can also be replaced by $\max_{S,A}\|\widehat{M}(S,A) - M^\star(S,A)\|$ (see the proof that is deferred to Section C.). The dependency on $\kappa$ may not be tight for real-life instances, but we note that most analysis of similar nature loses the additional $\kappa$ factor Schulman et al. (2015a); Achiam et al. (2017), and it's inevitable in the worst-case.

**A telescoping lemma.**    Towards proving Propositions 4.2 and deriving stronger discrepancy bound, we define the following quantity that captures the discrepancy between $\widehat{M}$ and $M^\star$ on a single state-action pair $(s, a)$.

$$G^{\pi,\widehat{M}}(s,a) = \underset{\hat{s}'\sim\widehat{M}(\cdot|s,a)}{\mathbb{E}} V^{\pi,\widehat{M}}(\hat{s}') - \underset{s'\sim M^\star(\cdot|s,a)}{\mathbb{E}} V^{\pi,\widehat{M}}(s') \tag{4.5}$$

Note that if $M, \widehat{M}$ are deterministic, then $G^{\pi,\widehat{M}}(s,a) = V^{\pi,\widehat{M}}(\widehat{M}(s,a)) - V^{\pi,\widehat{M}}(M^\star(s,a))$. We give a telescoping lemma that decompose the discrepancy between $V^{\pi,M}$ and $V^{\pi,M^\star}$ into the expected single-step discrepancy $G$.

**Lemma 4.3.** *[Telescoping Lemma] Recall that $\kappa := \gamma(1-\gamma)^{-1}$. For any policy $\pi$ and dynamical models $M, \widehat{M}$, we have that*

$$V^{\pi,\widehat{M}} - V^{\pi,M} = \kappa \underset{\substack{S\sim\rho^{\pi,M} \\ A\sim\pi(\cdot|S)}}{\mathbb{E}} \left[ G^{\pi,\widehat{M}}(S,A) \right] \tag{4.6}$$

The proof is reminiscent of the telescoping expansion in Kakade & Langford (2002) (c.f. Schulman et al. (2015a)) for characterizing the value difference of two policies, but we apply it to deal with the discrepancy between models. The detail is deferred to Section B. With the telescoping Lemma 4.3, Proposition 4.1 follows straightforwardly from Lipschitzness of the imaginary value function. Proposition 4.2 follows from that $\rho^\pi$ and $\rho^{\pi_{\text{ref}}}$ are close. We defer the proof to Appendix C.

## 4.2 REPRESENTATION-INVARIANT DISCREPANCY BOUNDS

The main limitation of the norm-based discrepancy bounds in previous subsection is that it depends on the state representation. Let $\mathcal{T}$ be a one-to-one map from the state space $\mathcal{S}$ to some other space $\mathcal{S}'$,

and for simplicity of this discussion let's assume a model $M$ is deterministic. Then if we represent every state $s$ by its transformed representation $\mathcal{T}s$, then the transformed model $M^{\mathcal{T}}$ defined as $M^{\mathcal{T}}(s,a) \triangleq \mathcal{T}M(\mathcal{T}^{-1}s, a)$ together with the transformed reward $R^{\mathcal{T}}(s,a) \triangleq R(\mathcal{T}^{-1}s, a)$ and transformed policy $\pi^{\mathcal{T}}(s) \triangleq \pi(\mathcal{T}^{-1}s)$ is equivalent to the original set of the model, reward, and policy in terms of the performance (Lemma C.1). Thus such transformation $\mathcal{T}$ is not identifiable from only observing the reward. However, the norm in the state space is a notion that depends on the hidden choice of the transformation $\mathcal{T}$. [5]

Another limitation is that the loss for the model learning should also depend on the state itself instead of only on the difference $\widehat{M}(S,A) - M^{\star}(S,A)$. It is possible that when $S$ is at a critical position, the prediction error needs to be highly accurate so that the model $\widehat{M}$ can be useful for planning. On the other hand, at other states, the dynamical model is allowed to make bigger mistakes because they are not essential to the reward.

We propose the following discrepancy bound towards addressing the limitations above. Recall the definition of $G^{\pi,\widehat{M}}(s,a) = V^{\pi,\widehat{M}}(\widehat{M}(s,a)) - V^{\pi,\widehat{M}}(M^{\star}(s,a))$ which measures the difference between $\widehat{M}(s,a))$ and $M^{\star}(s,a)$ according to their imaginary rewards. We construct a discrepancy bound using the absolute value of $G$. Let's define $\varepsilon_1$ and $\varepsilon_{\max}$ as the average of $|G^{\pi,\hat{M}}|$ and its maximum: $\varepsilon_1 = \mathbb{E}_{S \sim \rho^{\pi_{\text{ref}}}} \left[ \left| G^{\pi,\widehat{M}}(S,A) \right| \right]$ and $\varepsilon_{\max} = \max_S \left| G^{\pi,\widehat{M}}(S) \right|$ where $G^{\pi,\widehat{M}}(S) = \mathbb{E}_{A \sim \pi} \left[ G^{\pi,\widehat{M}}(S,A) \right]$. We will show that the following discrepancy bound $D^G_{\pi_{\text{ref}}}(\widehat{M}, \pi)$ satisfies the property (R1), (R2).

$$D^G_{\pi_{\text{ref}}}(\widehat{M}, \pi) = \kappa \cdot \varepsilon_1 + \kappa^2 \delta \varepsilon_{\max} \tag{4.7}$$

**Proposition 4.4.** *Let $d^{\text{KL}}$ and $D^G$ be defined as in equation* (4.3) *and* (4.7). *Then the choice $d = d^{\text{KL}}$ and $D = D^G$ satisfies the basic requirements (equation* (R1) *and* (R2)). *Moreover, $G$ is invariant w.r.t any one-to-one transformation of the state space (in the sense of equation C.1 in the proof).*

The proof follows from the telescoping lemma (Lemma 4.3) and is deferred to Section C. We remark that the first term $\kappa \varepsilon_1$ can in principle be estimated and optimized approximately: the expectation be replaced by empirical samples from $\rho^{\pi_{\text{ref}}}$, and $G^{\pi,\hat{M}}$ is an analytical function of $\pi$ and $\widehat{M}$ when they are both deterministic, and therefore can be optimized by back-propagation through time (BPTT). (When $\pi$ and $\widehat{M}$ and are stochastic with a re-parameterizable noise such as Gaussian distribution Kingma & Welling (2013), we can also use back-propagation to estimate the gradient.) The second term in equation (4.7) is difficult to optimize because it involves the maximum. However, it can be in theory considered as a second-order term because $\delta$ can be chosen to be a fairly small number. (In the refined bound in Section A, the dependency on $\delta$ is even milder.)

*Remark* 4.5. Proposition 4.4 intuitively suggests a technical reason of why model-based approach can be more sample-efficient than policy gradient based algorithms such as TRPO or PPO (Schulman et al., 2015a; 2017). The approximation error of $V^{\pi,\widehat{M}}$ in model-based approach decreases as the model error $\varepsilon_1, \varepsilon_{\max}$ decrease or the neighborhood size $\delta$ decreases, whereas the approximation error in policy gradient only linearly depends on the the neighborhood size Schulman et al. (2015a). In other words, model-based algorithms can trade model accuracy for a larger neighborhood size, and therefore the convergence can be faster (in terms of outer iterations.) This is consistent with our empirical observation that the model can be accurate in a descent neighborhood of the current policy so that the constraint (3.4) can be empirically dropped. We also refine our bonds in Section A, where the discrepancy bounds is proved to decay faster in $\delta$.

## 5 ADDITIONAL RELATED WORK

Model-based reinforcement learning is expected to require fewer samples than model-free algorithms (Deisenroth et al., 2013) and has been successfully applied to robotics in both simulation and in the real world (Deisenroth & Rasmussen, 2011b; Morimoto & Atkeson, 2003; Deisenroth et al., 2011) using dynamical models ranging from Gaussian process (Deisenroth & Rasmussen, 2011b; Ko

---

[5]That said, in many cases the reward function itself is known, and the states have physical meanings, and therefore we may be able to use the domain knowledge to figure out the best norm.

& Fox, 2009), time-varying linear models (Levine & Koltun, 2013; Lioutikov et al., 2014; Levine & Abbeel, 2014; Yip & Camarillo, 2014), mixture of Gaussians (Khansari-Zadeh & Billard, 2011), to neural networks (Hunt et al., 1992; Nagabandi et al., 2017; Kurutach et al., 2018; Tangkaratt et al., 2014; Sanchez-Gonzalez et al., 2018; Pascanu et al., 2017). In particular, the work of Kurutach et al. (2018) uses an ensemble of neural networks to learn the dynamical model, and significantly reduces the sample complexity compared to model-free approaches. The work of Chua et al. (2018) makes further improvement by using a probabilistic model ensemble. Clavera et al. (Clavera et al., 2018) extended this method with meta-policy optimization and improve the robustness to model error. In contrast, we focus on theoretical understanding of model-based RL and the design of new algorithms, and our experiments use a *single* neural network to estimate the dynamical model.

Our discrepancy bound in Section 4 is closely related to the work (Farahmand et al., 2017) on the value-aware model loss. Our approach differs from it in three details: a) we use the absolute value of the value difference instead of the squared difference; b) we use the imaginary value function from the estimated dynamical model to define the loss, which makes the loss purely a function of the estimated model and the policy; c) we show that the iterative algorithm, using the loss function as a building block, can converge to a local maximum, partly by cause of the particular choices made in a) and b). Asadi et al. (2018) also study the discrepancy bounds under Lipschitz condition of the MDP.

Prior work explores a variety of ways of combining model-free and model-based ideas to achieve the best of the two methods (Sutton, 1991; 1990; Racanière et al., 2017; Mordatch et al., 2016; Sun et al., 2018). For example, estimated models (Levine & Koltun, 2013; Gu et al., 2016; Kalweit & Boedecker, 2017) are used to enrich the replay buffer in the model-free off-policy RL. Pong et al. (2018b) proposes goal-conditioned value functions trained by model-free algorithms and uses it for model-based controls. Feinberg et al. (2018); Buckman et al. (2018) use dynamical models to improve the estimation of the value functions in the model-free algorithms.

On the control theory side, Dean et al. (2018; 2017) provide strong finite sample complexity bounds for solving linear quadratic regulator using model-based approach. Boczar et al. (2018) provide finite-data guarantees for the "coarse-ID control" pipeline, which is composed of a system identification step followed by a robust controller synthesis procedure. Our method is inspired by the general idea of maximizing a low bound of the reward in (Dean et al., 2017). By contrast, our work applies to non-linear dynamical systems. Our algorithms also estimate the models iteratively based on trajectory samples from the learned policies.

Strong model-based and model-free sample complexity bounds have been achieved in the tabular case (finite state space). We refer the readers to (Kakade et al., 2018; Dann et al., 2017; Szita & Szepesvári, 2010; Kearns & Singh, 2002; Jaksch et al., 2010; Agrawal & Jia, 2017) and the reference therein. Our work focus on continuous and high-dimensional state space (though the results also apply to tabular case).

Another line of work of model-based reinforcement learning is to learn a dynamic model in a hidden representation space, which is especially necessary for pixel state spaces (Kakade et al., 2018; Dann et al., 2017; Szita & Szepesvári, 2010; Kearns & Singh, 2002; Jaksch et al., 2010). Srinivas et al. (2018) shows the possibility to learn an abstract transition model to imitate expert policy. Oh et al. (2017) learns the hidden state of a dynamical model to predict the value of the future states and applies RL or planning on top of it. Serban et al. (2018); Ha & Schmidhuber (2018) learns a bottleneck representation of the states. Our framework can be potentially combined with this line of research.

# 6 PRACTICAL IMPLEMENTATION AND EXPERIMENTS

## 6.1 PRACTICAL IMPLEMENTATION

We design with simplification of our framework a variant of model-based RL algorithms, Stochastic Lower Bound Optimization (SLBO). First, we removed the constraints (3.4). Second, we stop the gradient w.r.t $M$ (but not $\pi$) from the occurrence of $M$ in $V^{\pi,M}$ in equation (3.3) (and thus our practical implementation is not optimism-driven.)

Extending the discrepancy bound in Section 4.1, we use a multi-step prediction loss for learning the models with $\ell_2$ norm. For a state $s_t$ and action sequence $a_{t:t+h}$, we define the $h$-step prediction $\hat{s}_{t+h}$

as $\hat{s}_t = s_t$, and for $h \geq 0$, $\hat{s}_{t+h+1} = \widehat{M}_\phi(\hat{s}_{t+h}, a_{t+h})$, The *H-step loss* is then defined as

$$\mathcal{L}_\phi^{(H)}((s_{t:t+h}, a_{t:t+h}); \phi) = \frac{1}{H} \sum_{i=1}^{H} \|(\hat{s}_{t+i} - \hat{s}_{t+i-1}) - (s_{t+i} - s_{t+i-1})\|_2. \quad (6.1)$$

A similar loss is also used in Nagabandi et al. (2017) for validation. We note that motivation by the theory in Section 4.1, we use $\ell_2$-norm instead of the square of $\ell_2$ norm. The loss function we attempt to optimize at iteration $k$ is thus[6]

$$\max_{\phi, \theta} \ V^{\pi_\theta, \text{sg}(\widehat{M}_\phi)} - \lambda \mathbb{E}_{(s_{t:t+h}, a_{t:t+h}) \sim \pi_k, M^\star} \left[ \mathcal{L}_\phi^{(H)}((s_{t:t+h}, a_{t:t+h}); \phi) \right] \quad (6.2)$$

where $\lambda$ is a tunable parameter and sg denotes the stop gradient operation.

We note that the term $V^{\pi_\theta, \text{sg}(\widehat{M}_\phi)}$ depends on both the parameter $\theta$ and the parameter $\phi$ but there is no gradient passed through $\phi$, whereas $\mathcal{L}_\phi^{(H)}$ only depends on the the $\phi$. We optimize equation (6.2) by alternatively maximizing $V^{\pi_\theta, \text{sg}(\widehat{M}_\phi)}$ and minimizing $\mathcal{L}_\phi^{(H)}$: for the former, we use TRPO with samples from the estimated dynamical model $\widehat{M}_\phi$ (by treating $\widehat{M}_\phi$ as a fixed simulator), and for the latter we use standard stochastic gradient methods. Algorithm 2 gives a pseudo-code for the algorithm. The $n_{\text{model}}$ and $n_{\text{policy}}$ iterations are used to balance the number of steps of TRPO and Adam updates within the loop indexed by $n_{\text{inner}}$.[7]

---

**Algorithm 2** Stochastic Lower Bound Optimization (SLBO)

---

1: Initialize model network parameters $\phi$ and policy network parameters $\theta$
2: Initialize dataset $\mathcal{D} \leftarrow \emptyset$
3: **for** $n_{\text{outer}}$ iterations **do**
4:      $\mathcal{D} \leftarrow \mathcal{D} \cup \{$ collect $n_{\text{collect}}$ samples from real environment using $\pi_\theta$ with noises $\}$
5:      **for** $n_{\text{inner}}$ iterations **do**                ▷ optimize (6.2) with stochastic alternating updates
6:          **for** $n_{\text{model}}$ iterations **do**
7:              optimize (6.1) over $\phi$ with sampled data from $\mathcal{D}$ by one step of Adam
8:          **for** $n_{\text{policy}}$ iterations **do**
9:              $\mathcal{D}' \leftarrow \{$ collect $n_{\text{trpo}}$ samples using $\widehat{M}_\phi$ as dynamics $\}$
10:             optimize $\pi_\theta$ by running TRPO on $\mathcal{D}'$

---

**Power of stochasticity and connection to standard MB RL:** We identify the main advantage of our algorithms over standard model-based RL algorithms is that we alternate the updates of the model and the policy within an outer iteration. By contrast, most of the existing model-based RL methods only optimize the models once (for a lot of steps) after collecting a batch of samples (see Algorithm 3 for an example). The stochasticity introduced from the alternation with stochastic samples seems to dramatically reduce the overfitting (of the policy to the estimated dynamical model) in a way similar to that SGD regularizes ordinary supervised training. [8] Another way to view the algorithm is that the model obtained from line 7 of Algorithm 2 at different inner iteration serves as an ensemble of models. We do believe that a cleaner and easier instantiation of our framework (with optimism) exists, and the current version, though performing very well, is not necessarily the best implementation.

**Entropy regularization:** An additional component we apply to SLBO is the commonly-adopted entropy regularization in policy gradient method (Williams & Peng, 1991; Mnih et al., 2016), which was found to significantly boost the performance in our experiments (ablation study in Appendix F.5). Specifically, an additional entropy term is added to the objective function in TRPO. We hypothesize that entropy bonus helps exploration, diversifies the collected data, and thus prevents overfitting.

---

[6]This is technically not a well-defined mathematical objective. The sg operation means identity when the function is evaluated, whereas when computing the update, $\text{sg}(M_\phi)$ is considered fixed.

[7]In principle, to balance the number of steps, it suffices to take one of $n_{\text{model}}$ and $n_{\text{policy}}$ to be 1. However, empirically we found the optimal balance is achieved with larger $n_{\text{model}}$ and $n_{\text{policy}}$, possibly due to complicated interactions between the two optimization problem.

[8]Similar stochasticity can potentially be obtained by an extreme hyperparameter choice of the standard MB RL algorithm: in each outer iteration of Algorithm 3, we only sample a very small number of trajectories and take a few model updates and policy updates. We argue our interpretation of stochastic optimization of the lower bound (6.2) is more natural in that it reveals the regularization from stochastic optimization.

## 6.2 EXPERIMENTAL RESULTS

We evaluate our algorithm SLBO (Algorithm 2) on five continuous control tasks from rllab (Duan et al., 2016), including Swimmer, Half Cheetah, Humanoid, Ant, Walker. All environments that we test have a maximum horizon of 500, which is longer than most of the existing model-based RL work (Nagabandi et al., 2017; Kurutach et al., 2018). (Environments with longer horizons are commonly harder to train.) More details can be found in Appendix F.1.

**Baselines.** We compare our algorithm with 3 other algorithms including: (1) Soft Actor-Critic (SAC) (Haarnoja et al., 2018), the state-of-the-art model-free off-policy algorithm in sample efficiency; (2) Trust-Region Policy Optimization (TRPO) (Schulman et al., 2015a), a policy-gradient based algorithm; and (3) Model-Based TRPO, a standard model-based algorithm described in Algorithm 3. Details of these algorithms can be found in Appendix F.4.[9]

The result is shown in Figure 1. In Fig 1, our algorithm shows superior convergence rate (in number of samples) than all the baseline algorithms while achieving better final performance with 1M samples. Specifically, we mark model-free TRPO performance after 8 million steps by the dotted line in Fig 1 and find out that our algorithm can achieve comparable or better final performance in one million steps. For ablation study, we also add the performance of SLBO-MSE, which corresponds to running SLBO with squared $\ell_2$ model loss instead of $\ell_2$. SLBO-MSE performs significantly worse than SLBO on four environments, which is consistent with our derived model loss in Section 4.1. We also study the performance of SLBO and baselines with 4 million training samples in F.5. Ablation study of multi-step model training can be found in Appendix F.5.[10]

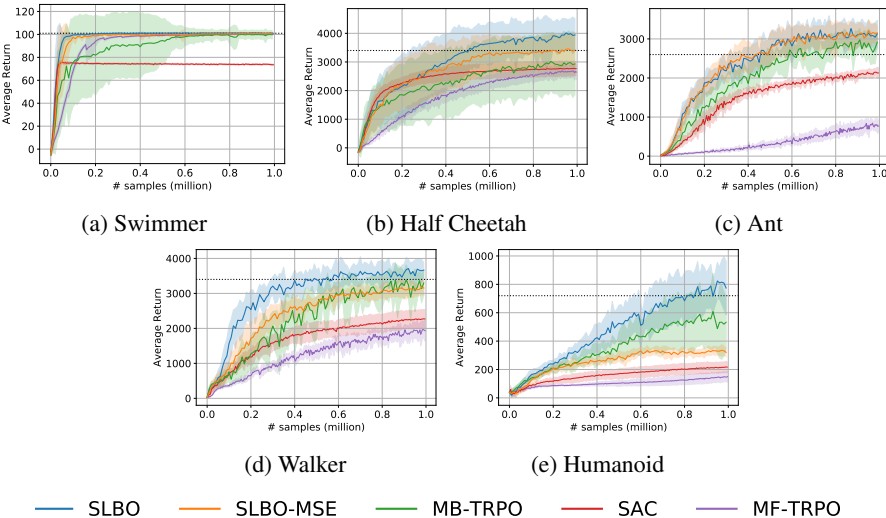

Figure 1: Comparison between SLBO (ours), SLBO with squared $\ell^2$ model loss (SLBO-MSE), vanilla model-based TRPO (MB-TRPO), model-free TRPO (MF-TRPO), and Soft Actor-Critic (SAC). We average the results over 10 different random seeds, where the solid lines indicate the mean and shaded areas indicate one standard deviation. The dotted reference lines are the total rewards of MF-TRPO after 8 million steps.

## 7 CONCLUSIONS

We devise a novel algorithmic framework for designing and analyzing model-based RL algorithms with the guarantee to convergence monotonically to a local maximum of the reward. Experimental results show that our proposed algorithm (SLBO) achieves new state-of-the-art performance on several mujoco benchmark tasks when one million or fewer samples are permitted.

---

[9]We did not have the chance to implement the competitive random search algorithms in (Mania et al., 2018) yet, although our test performance with 500 episode length is higher than theirs with 1000 episode on Half Cheetach (3950 by ours vs 2345 by theirs) and Walker (3650 by ours vs 894 by theirs).

[10]Videos demonstrations are available at https://sites.google.com/view/algombrl/home. A link to the codebase is available at https://github.com/roosephu/slbo.

A compelling (but obvious) empirical open question then given rise to is whether model-based RL can achieve near-optimal reward on other more complicated tasks or real-world robotic tasks with fewer samples. We believe that understanding the trade-off between optimism and robustness is essential to design more sample-efficient algorithms. Currently, we observed empirically that the optimism-driven part of our proposed meta-algorithm (optimizing $V^{\pi,\widehat{M}}$ over $\widehat{M}$) may lead to instability in the optimization, and therefore don't in general help the performance. It's left for future work to find practical implementation of the optimism-driven approach.

In our theory, we assume that the parameterized model class contains the true dynamical model. Removing this assumption is also another interesting open question. It would be also very interesting if the theoretical analysis can be applied other settings involving model-based approaches (e.g., model-based imitation learning).

**Acknowledgments:**

We thank the anonymous reviewers for detailed, thoughtful, and helpful reviews. We'd like to thank Emma Brunskill, Chelsea Finn, Shane Gu, Ben Recht, and Haoran Tang for many helpful comments and discussions.

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

## A    REFINED BOUNDS

The theoretical limitation of the discrepancy bound $D^G(\widehat{M}, \pi)$ is that the second term involving $\varepsilon_{\max}$ is not rigorously optimizable by stochastic samples. In the worst case, there seem to exist situations where such infinity norm of $G^{\pi, \widehat{M}}$ is inevitable. In this section we tighten the discrepancy bounds with a different closeness measure $d$, $\chi^2$-divergence, in the policy space, and the dependency on the $\varepsilon_{\max}$ is smaller (though not entirely removed.) We note that $\chi^2$-divergence has the same second order approximation as KL-divergence around the local neighborhood the reference policy and thus locally affects the optimization much.

We start by defining a re-weighted version $\beta^{\pi}$ of the distribution $\rho^{\pi}$ where examples in later step are slightly weighted up. We can effectively sample from $\beta^{\pi}$ by importance sampling from $\rho^{\pi}$

**Definition A.1.** For a policy $\pi$, define $\beta^{\pi}$ as the *re-weighted* version of discounted distribution of the states visited by $\pi$ on $M^{\star}$. Recall that $p_{S_t^{\pi}}$ is the distribution of the state at step $t$, we define $\beta^{\pi} = (1 - \gamma)^2 \sum_{t=1}^{\infty} t\gamma^{t-1} p_{S_t^{\pi}}$.

Then we are ready to state our discrepancy bound. Let

$$d^{\chi^2}(\pi, \pi_{\text{ref}}) = \max\{ \mathop{\mathbb{E}}_{S \sim \rho^{\pi_{\text{ref}}}} \left[ \chi^2(\pi(\cdot|S), \pi_{\text{ref}}(\cdot|S)) \right] , \mathop{\mathbb{E}}_{S \sim \beta^{\pi_{\text{ref}}}} \left[ \chi^2(\pi(\cdot|S), \pi_{\text{ref}}(\cdot|S)) \right] \} \tag{A.1}$$

$$D_{\pi_{\text{ref}}}^{\chi^2}(\widehat{M}, \pi) = (1 - \gamma)^{-1}\varepsilon_1 + (1 - \gamma)^{-2}\delta\varepsilon_2 + (1 - \gamma)^{-5/2}\delta^{3/2}\varepsilon_{\max} \tag{A.2}$$

where $\varepsilon_2 = \mathbb{E}_{S \sim \beta^{\pi_{\text{ref}}}} \left[ G^{\pi, \widehat{M}}(S, A)^2 \right]$ and $\varepsilon_1, \varepsilon_{\max}$ are defined in equation (4.2).

**Proposition A.2.** *The discrepancy bound $D^{\chi^2}$ and closeness measure $d^{\chi^2}$ satisfies requirements* (R1) *and* (R2).

We defer the proof to Section C so that we can group relevant proofs with similar tools together. Some of these tools may be of independent interests and used for better analysis of model-free reinforcement learning algorithms such as TRPO Schulman et al. (2015a), PPO Schulman et al. (2017) and CPO Achiam et al. (2017).

## B    PROOF OF LEMMA 4.3

*Proof of Lemma 4.3.* Let $W_j$ be the cumulative reward when we use dynamical model $M$ for $j$ steps and then $\widehat{M}$ for the rest of the steps, that is,

$$W_j = \mathop{\mathbb{E}}_{\substack{\forall t \geq 0, A_t \sim \pi(\cdot|S_t) \\ \forall j > t \geq 0, S_{t+1} \sim M(\cdot|S_t, A_t) \\ \forall t \geq j, S_{t+1} \sim \widehat{M}(\cdot|S_t, A_t)}} \left[ \sum_{t=0}^{\infty} \gamma^t R(S_t, A_t) \mid S_0 = s \right]$$

By definition, we have that $W_{\infty} = V^{\pi, M}(s)$ and $W_0 = V^{\pi, \widehat{M}}(s)$. Then, we decompose the target into a telescoping sum,

$$V^{\pi, M}(s) - V^{\pi, \widehat{M}}(s) = \sum_{j=0}^{\infty} (W_{j+1} - W_j) \tag{B.1}$$

Now we re-write each of the summands $W_{j+1} - W_j$. Comparing the trajectory distribution in the definition of $W_{j+1}$ and $W_j$, we see that they only differ in the dynamical model applied in $j$-th step. Concretely, $W_j$ and $W_{j+1}$ can be rewritten as $W_j = R + \mathbb{E}_{S_j, A_j \sim \pi, M} \left[ \mathbb{E}_{\widehat{S}_{j+1} \sim \widehat{M}(\cdot|S_j, A_j)} \left[ \gamma^{j+1} V^{\pi, \widehat{M}}(\widehat{S}_{j+1}) \right] \right]$ and $W_{j+1} = R + \mathbb{E}_{S_j, A_j \sim \pi, M^{\star}} \left[ \mathbb{E}_{S_{j+1} \sim M(\cdot|S_j, A_j)} \left[ \gamma^{j+1} V^{\pi, \widehat{M}}(S_{j+1}) \right] \right]$ where $R$ denotes the reward from the first $j$ steps from policy $\pi$ and model $M^{\star}$. Canceling the shared term in the two equations above, we get

$$W_{j+1} - W_j = \gamma^{j+1} \mathop{\mathbb{E}}_{S_j, A_j \sim \pi, M} \left[ \mathop{\mathbb{E}}_{\substack{\widehat{S}_{j+1} \sim \widehat{M}(\cdot|S_j, A_j) \\ S_{j+1} \sim M(\cdot|S_j, A_j)}} \left[ V^{\pi, \widehat{M}}(S_{j+1}) - V^{\pi, \widehat{M}}(\widehat{S}_{j+1}) \right] \right]$$

Combining the equation above with equation (B.1) concludes that

$$V^{\pi,M} - V^{\pi,\widehat{M}} = \frac{\gamma}{1-\gamma} \underset{S\sim\rho^\pi, A\sim\pi(S)}{\mathbb{E}} \left[ \underset{S'\sim M^\star(\cdot|S,A)}{\mathbb{E}} V^{\pi,\widehat{M}}(S') - \underset{\hat{S}'\sim\widehat{M}(\cdot|S,A)}{\mathbb{E}} V^{\pi,\widehat{M}}(\hat{S}') \right]$$

$\square$

# C  MISSING PROOFS IN SECTION 4

## C.1  PROOF OF PROPOSITION 4.4

Towards proving the second part of Proposition 4.4 regarding the invariance, we state the following lemma:

**Lemma C.1.** *Suppose for simplicity the model and the policy are both deterministic. For any one-to-one transformation from $\mathcal{S}$ to $\mathcal{S}'$, let $M^\mathcal{T}(s,a) \triangleq \mathcal{T}M(\mathcal{T}^{-1}s,a)$, $R^\mathcal{T}(s,a) \triangleq R(\mathcal{T}^{-1}s,a)$, and $\pi^\mathcal{T}(s) \triangleq \pi(\mathcal{T}^{-1}s)$ be a set of transformed model, reward and policy. Then we have that $(M,\pi,R)$ is equivalent to $(M^\mathcal{T}, \pi^\mathcal{T}, R^\mathcal{T})$ in the sense that*

$$V^{\pi^\mathcal{T}, M^\mathcal{T}}(\mathcal{T}s) = V^{\pi,M}(s)$$

*where the value function $V^{\pi^\mathcal{T}, M^\mathcal{T}}$ is defined with respect to $R^\mathcal{T}$.*

*Proof.* Let $s_0^\mathcal{T} = \mathcal{T}s, \ldots,$ be the sequence of states visited by policy $\pi^\mathcal{T}$ on model $M^\mathcal{T}$ starting from $s$. We have that $s_0^\mathcal{T} = \mathcal{T}s = \mathcal{T}s_0$. We prove by induction that $s_t^\mathcal{T} = \mathcal{T}s_t$. Assume this is true for some value $t$, then we prove that $s_{t+1}^\mathcal{T} = \mathcal{T}s_{t+1}$ holds:

$$\begin{aligned}
s_{t+1}^\mathcal{T} &= M^\mathcal{T}(s_t^\mathcal{T}, \pi^\mathcal{T}(s_t^\mathcal{T})) = M^\mathcal{T}(\mathcal{T}s_t, \pi^\mathcal{T}(\mathcal{T}s_t)) && \text{(by inductive hypothesis)} \\
&= \mathcal{T}M(s_t, \pi(s_t)) && \text{(by defintion of } M^\mathcal{T}, \pi^\mathcal{T}) \\
&= \mathcal{T}s_{t+1}
\end{aligned}$$

Thus we have $R^\mathcal{T}(s_t^\mathcal{T}, a_t^\mathcal{T}) = R(s_t, a_t)$. Therefore $V^{\pi^\mathcal{T}, M^\mathcal{T}}(\mathcal{T}s) = V^{\pi,M}(s)$.  $\square$

*Proof of Proposition 4.4.* We first show the invariant of $G$ under deterministic models and policies. The same result applies to stochastic policies with slight modification. Let $s^\mathcal{T} = \mathcal{T}s$. We consider the transformation applied to $M$ and $M^\star$ and the resulting $G$ function

$$G^\mathcal{T}(s^\mathcal{T}, a) \triangleq |V^{\pi^\mathcal{T}, M^\mathcal{T}}(M^\mathcal{T}(s^\mathcal{T}, a)) - V^{\pi^\mathcal{T}, M^\mathcal{T}}(M^{\star,\mathcal{T}}(s^\mathcal{T}, a))|$$

Note that by Lemma C.1, we have that $V^{\pi^\mathcal{T}, M^\mathcal{T}}(M^\mathcal{T}(s^\mathcal{T}, a)) = V^{\pi,M}(\mathcal{T}^{-1}M^\mathcal{T}(s^\mathcal{T}, a)) = V^{\pi,M}(M(s,a))$. Similarly, $V^{\pi^\mathcal{T}, M^\mathcal{T}}(M^{\star,\mathcal{T}}(s^\mathcal{T}, a)) = V^{\pi,M}(M^\star(s,a))$. Therefore we obtain that

$$G^\mathcal{T}(s^\mathcal{T}, a) = G(s, a) \tag{C.1}$$

By Lemma 4.3 and triangle inequality, we have that

$$\frac{1-\gamma}{\gamma} \left| V^{\pi,M} - V^{\pi,\widehat{M}} \right| \leq \underset{S\sim\rho^\pi}{\mathbb{E}} \left[ \left| G^{\pi,\widehat{M}}(S) \right| \right] \qquad\qquad \text{(triangle inequality)}$$

$$\leq \underset{S\sim\rho^{\pi_{\text{ref}}}}{\mathbb{E}} \left[ \left| G^{\pi,\widehat{M}}(S) \right| \right] + |\rho^\pi - \rho^{\pi_{\text{ref}}}|_1 \cdot \max_S \left| G^{\pi,\widehat{M}}(S) \right|$$

$$\text{(Holder inequality)}$$

By Corollary E.7 we have that $|\rho^\pi - \rho^{\pi_{\text{ref}}}|_1 \leq \frac{\gamma}{1-\gamma} \mathbb{E}_{S\sim\rho^{\pi_{\text{ref}}}} \left[ KL(\pi(S), \pi_{\text{ref}}(S))^{1/2}|S \right] = \frac{\delta\gamma}{1-\gamma}$. Combining this with the equation above, we complete the proof.  $\square$

## C.2  PROOF OF PROOF OF PROPOSITION A.2

*Proof of Proposition A.2.* Let $\mu$ be the distribution of the initial state $S_0$, and let $P'$ and $P$ be the state-to-state transition kernel under policy $\pi$ and $\pi_{\text{ref}}$. Let $\bar{G} = (1 - \gamma) \sum_{k=0}^{\infty} \gamma^k P^k$ and $\bar{G}' = (1 - \gamma) \sum_{k=0}^{\infty} \gamma^k P'^k$. Under these notations, we can re-write $\rho^{\pi_{\text{ref}}} = \bar{G}\mu$ and $\rho^\pi = \bar{G}'\mu$. Moreover, we observe that $\beta^{\pi_{\text{ref}}} = \bar{G} P \bar{G}\mu$.

Let $\delta_1 = (1 - \gamma)^{-1} \chi^2_{\bar{G}\mu}(P', P)^{1/2}$ and $\delta_2 = (1 - \gamma)^{-1} \chi^2_{\bar{G}P\bar{G}\mu}(P', P)^{1/2}$ by the $\chi^2$ divergence between $P'$ and $P$, measured with respect to distributions $\bar{G}\mu = \rho^{\pi_{\text{ref}}}$ and $\bar{G} P \bar{G}\mu = \beta^{\pi_{\text{ref}}}$. By Lemma D.1, we have that the $\chi^2$-divergence between the states can be bounded by the $\chi^2$-divergence between the actions in the sense that:

$$\chi^2_{\bar{G}\mu}(P', P)^{1/2} = \chi^2_{\rho^{\pi_{\text{ref}}}}(P', P)^{1/2} \leq \mathop{\mathbb{E}}_{S \sim \rho^{\pi_{\text{ref}}}} \left[ \chi^2(\pi(\cdot|S), \pi_{\text{ref}}(\cdot|S)) \right]^{1/2}$$

$$\chi^2_{\bar{G}P\bar{G}\mu}(P', P)^{1/2} = \chi^2_{\beta^{\pi_{\text{ref}}}}(P', P)^{1/2} \leq \mathop{\mathbb{E}}_{S \sim \beta^{\pi_{\text{ref}}}} \left[ \chi^2(\pi(\cdot|S), \pi_{\text{ref}}(\cdot|S)) \right]^{1/2}$$

Therefore we obtain that $\delta_1 \leq (1 - \gamma)^{-1}\delta$, $\delta_2 \leq (1 - \gamma)^{-1}\delta$. Let $f(s) = G^{\pi, \widehat{M}}(s)$. By Lemma D.4, we can control the difference between $\langle \rho^{\pi_{\text{ref}}}, f \rangle$ and $\langle \rho^\pi, f \rangle$ by

$$\left| \mathop{\mathbb{E}}_{S \sim \rho^{\text{ref}}} \left[ G^{\pi, \widehat{M}}(S) \right] - \mathop{\mathbb{E}}_{S \sim \rho^\pi} \left[ G^{\pi, \widehat{M}}(S) \right] \right| = |\langle \rho^{\pi_{\text{ref}}}, f \rangle - \langle \rho^\pi, f \rangle|$$

$$\leq \delta_1 \langle \bar{G} P \bar{G}\mu, f^2 \rangle^{1/2} + \delta_1 \delta_2^{1/2} \|f\|_\infty$$

$$\leq \delta_1 \varepsilon_2 + \delta_1 \delta_2^{1/2} \varepsilon_{\max}$$

It follows that

$$\left| V^{\pi, \widehat{M}} - V^{\pi, M} \right| \leq \gamma(1 - \gamma)^{-1} \left| \mathop{\mathbb{E}}_{S \sim \rho^\pi} \left[ G^{\pi, \widehat{M}}(S) \right] \right| \qquad \text{(by Lemma 4.3)}$$

$$\leq \gamma(1 - \gamma)^{-1} \left( \left| \mathop{\mathbb{E}}_{S \sim \rho^{\text{ref}}} \left[ G^{\pi, \widehat{M}}(S) \right] \right| + \left| \mathop{\mathbb{E}}_{S \sim \rho^{\text{ref}}} \left[ G^{\pi, \widehat{M}}(S) - \mathop{\mathbb{E}}_{S \sim \rho^\pi} \left[ G^{\pi, \widehat{M}}(S) \right] \right] \right| \right)$$

$$\leq \gamma(1 - \gamma)^{-1} \left| \mathop{\mathbb{E}}_{S \sim \rho^{\text{ref}}} \left[ G^{\pi, \widehat{M}}(S) \right] \right| + \gamma(1 - \gamma)^{-1}\delta_1 \varepsilon_2 + \gamma(1 - \gamma)^{-1}\delta_1 \delta_2^{1/2} \varepsilon_{\max}$$

$$\leq (1 - \gamma)^{-1}\varepsilon_1 + (1 - \gamma)^{-2}\delta\varepsilon_2 + (1 - \gamma)^{-5/2}\delta^{3/2}\varepsilon_{\max}$$

$\square$

## C.3  PROOF OF PROPOSITION 4.1 AND 4.2

*Proof of Proposition 4.1 and 4.2 .* By definition of $G$ and the Lipschitzness of $V^{\pi, \widehat{M}}$, we have that $|G^{\pi, \widehat{M}}(s, a)| \leq L|\widehat{M}(s, a) - M^\star(s, a)|$. Then, by Lemma 4.3 and triangle inequality, we have that

$$\left| V^{\pi, \widehat{M}} - V^{\pi, M^\star} \right| = \kappa \cdot \left| \mathop{\mathbb{E}}_{\substack{S \sim \rho^{\pi, M} \\ A \sim \pi(\cdot|S)}} \left[ G^{\pi, \widehat{M}}(S, A) \right] \right| \leq \kappa \mathop{\mathbb{E}}_{\substack{S \sim \rho^{\pi, M} \\ A \sim \pi(\cdot|S)}} \left[ \left| G^{\pi, \widehat{M}}(S, A) \right| \right]$$

$$\leq \kappa \mathop{\mathbb{E}}_{\substack{S \sim \rho^\pi \\ A \sim \pi(\cdot|S)}} \left[ \|\widehat{M}(S, A) - M^\star(S, A)\| \right] . \qquad (C.2)$$

Next we prove the main part of the proposition. Thus we proved Proposition 4.1. Note that for any distribution $\rho$ and $\rho'$ and function $f$, we have $\mathbb{E}_{S \sim \rho} f(S) = \mathbb{E}_{S \sim \rho'} f(S) + \langle \rho - \rho', f \rangle \leq \mathbb{E}_{S \sim \rho'} f(S) + \|\rho - \rho'\|_1 \|f\|_\infty$. Thus applying this inequality with $f(S) = \mathbb{E}_{A \sim \pi(\cdot|S)} \left[ \|\widehat{M}(S, A) - M^\star(S, A)\| \right]$, we obtain that

$$\mathop{\mathbb{E}}_{\substack{S \sim \rho^\pi \\ A \sim \pi(\cdot|S)}} \left[ \|\widehat{M}(S, A) - M^\star(S, A)\| \right] \leq \mathop{\mathbb{E}}_{\substack{S \sim \rho^{\pi_{\text{ref}}} \\ A \sim \pi(\cdot|S)}} \left[ \|\widehat{M}(S, A) - M^\star(S, A)\| \right]$$

$$+ \|\rho^{\pi_{\text{ref}}} - \rho\|_1 \max_S \mathop{\mathbb{E}}_{A \sim \pi(\cdot|S)} \left[ \|\widehat{M}(S, A) - M^\star(S, A)\| \right]$$

$$\leq \mathop{\mathbb{E}}_{\substack{S \sim \rho^{\pi_{\text{ref}}} \\ A \sim \pi(\cdot|S)}} \left[ \| \widehat{M}(S,A) - M^{\star}(S,A) \| \right] + 2\delta\kappa B \qquad (\text{C.3})$$

where the last inequality uses the inequalities (see Corollary E.7) that $\|\rho^{\pi} - \rho^{\pi_{\text{ref}}}\|_1 \leq \frac{\gamma}{1-\gamma} \mathbb{E}_{S \sim \rho^{\pi_{\text{ref}}}} \left[ KL(\pi(S), \pi_{\text{ref}}(S))^{1/2} | S \right] = \delta\kappa$ and that $\|\widehat{M}(S,A) - M^{\star}(S,A)\| \leq 2B$. Combining (C.3) and (C.2) we complete the proof of Proposition 4.2. $\qquad \square$

# D $\quad \chi^2$-DIVERGENCE BASED INEQUALITIES

**Lemma D.1.** *Let $S$ be a random variable over the domain $\mathcal{S}$. Let $\pi$ and $\pi'$ be two policies and and $A \sim \pi(\cdot \mid S)$ and $A' \sim \pi'(\cdot \mid S)$. Let $Y \sim M(\cdot \mid S, A)$ and $Y' \sim M(\cdot \mid S, A')$ be the random variables for the next states under two policies. Then,*

$$\mathbb{E}\left[\chi^2(Y|S, Y'|S)\right] \leq \mathbb{E}\left[\chi^2(A|S, A'|S)\right]$$

*Proof.* By definition, we have that $Y|S = s, A = a$ has the same density as $Y'|S = s, A' = a$ for any $a$ and $s$. Therefore by Theorem E.4 (setting $X, X', Y, Y'$ in Theorem E.4 by $A|S = s, A'|S = s, Y|S = s, Y'|S = s$ respectively), we have

$$\chi^2(Y|S = s, Y'|S = s) \leq \chi^2(A|S = s, A'|S = s)$$

Taking expectation over the randomness of $S$ we complete the proof.

$\qquad \square$

## D.1 PROPERTIES OF MARKOV PROCESSES

In this subsection, we consider bounded the difference of the distributions induced by two markov process starting from the same initial distributions $\mu$. Let $P, P'$ be two transition kernels. Let $G = \sum_{k=0}^{\infty} \gamma^k P^k$ and $\bar{G} = (1-\gamma)G$. Define $G'$ and $\bar{G}'$ similarly. Therefore we have that $\bar{G}\mu$ is the discounted distribution of states visited by the markov process starting from distribution $\mu$. In other words, if $\mu$ is the distribution of $S_0$, and $P$ is the transition kernel induced by some policy $\pi$, then $\bar{G}\mu = \rho^{\pi}$.

First of all, let $\Delta = \gamma(P' - P)$ and we note that with simple algebraic manipulation,

$$\bar{G}' - \bar{G} = (1-\gamma)^{-1}\bar{G}'\Delta\bar{G} \qquad (\text{D.1})$$

Let $f$ be some function. We will mostly interested in the difference between $\mathbb{E}_{S \sim \bar{G}\mu}[f]$ and $\mathbb{E}_{S \sim \bar{G}'\mu}[f]$, which can be rewritten as $\langle(\bar{G}' - \bar{G})\mu, f\rangle$. We will bound this quantity from above by some divergence measure between $P'$ and $P$.

We start off with a simple lemma that controls the form $\langle p - q, f\rangle$ by the $\chi^2$ divergence between $p$ and $q$. With this lemma we can reduce our problem of bounding $\langle(\bar{G}' - \bar{G})\mu, f\rangle$ to characterizing the $\chi^2$ divergence between $\bar{G}'\mu$ and $\bar{G}\mu$.

**Lemma D.2.** *Let $p$ and $q$ be probability distributions. Then we have*

$$\langle q - p, f\rangle^2 \leq \chi^2(q, p) \cdot \langle p, f^2\rangle$$

*Proof.* By Cauchy-Schwartz inequality, we have

$$\langle q - p, f\rangle^2 \leq \left(\int \frac{(q(x) - p(x))^2}{p(x)} dx\right)\left(\int p(x)f(x)^2\right) = \chi^2(q, p) \cdot \langle p, f^2\rangle$$

$\qquad \square$

The following Lemma is a refinement of the lemma above. It deals with the distributions $p$ and $q$ with the special structure $p = WP'\mu$ and $q = WP\mu$.

**Lemma D.3.** *Let $W, P', P$ be transition kernels and $\mu$ be a distribution. Then,*

$$\langle W(P'-P)\mu, f\rangle^2 \leq \chi^2_\mu(P', P)\langle WP\mu, f^2\rangle$$

*where $\chi^2_\mu(P', P)$ is a divergence between transitions defined in Definition E.3.*

*Proof.* By Lemma D.2 with $p = WP\mu$ and $q = WP'\mu$, we conclude that

$$\langle W(P'-P)\mu, f\rangle^2 \leq \chi^2(q, p) \cdot \langle p, f^2\rangle \leq \chi^2(WP'\mu, WP\mu)\langle WP\mu, f^2\rangle$$

By Theorem E.4 and Theorem E.5 we have that $\chi^2(WP'\mu, WP\mu) \leq \chi^2(P'\mu, P\mu) \leq \chi^2_\mu(P', P)$, plugging this into the equation above we complete the proof. $\qquad\square$

Now we are ready to state the main result of this subsection.

**Lemma D.4.** *Let $\bar{G}, \bar{G}', P', P, f$ as defined in the beginning of this section. Let $\delta_1 = (1 - \gamma)^{-1}\chi^2_{\bar{G}\mu}(P', P)^{1/2}$ and $\delta_2 = (1-\gamma)^{-1}\chi^2_{\bar{G}P\bar{G}\mu}(P', P)^{1/2}$. Then,*

$$\left|\langle \bar{G}'\mu, f\rangle - \langle \bar{G}\mu, f\rangle\right| \leq \delta_1 \|f\|_\infty \tag{D.2}$$

$$\left|\langle \bar{G}'\mu, f\rangle - \langle \bar{G}\mu, f\rangle\right| \leq \delta_1\langle \bar{G}P\bar{G}\mu, f^2\rangle^{1/2} + \delta_1\delta_2^{1/2}\|f\|_\infty$$

*Proof.* Recall by equation (D.1), we have

$$\langle(\bar{G}' - \bar{G})\mu, f\rangle = (1 - \gamma)^{-1}\langle \bar{G}'\Delta\bar{G}\mu, f\rangle \tag{D.3}$$

By Lemma D.3,

$$\langle \bar{G}'\Delta\bar{G}\mu, f\rangle \leq \chi^2_{\bar{G}\mu}(P', P)^{1/2}\langle \bar{G}'P\bar{G}\mu, f^2\rangle^{1/2} \tag{D.4}$$

By Holder inequality and the fact that $\|\bar{G}\|_{1\to 1} = 1$, $\|\bar{G}'\|_{1\to 1} = 1$ and $\|P\|_{1\to 1} = 1$, we have

$$\begin{aligned}
\langle \bar{G}'\Delta\bar{G}\mu, f\rangle &\leq \chi^2_{\bar{G}\mu}(P', P)^{1/2}\langle \bar{G}'P\bar{G}\mu, f^2\rangle^{1/2} \\
&\leq \chi^2_{\bar{G}\mu}(P', P)^{1/2}\|\bar{G}'P\bar{G}\mu\|_1^{1/2}\|f^2\|_\infty^{1/2} \\
&\leq \chi^2_{\bar{G}\mu}(P', P)^{1/2}\|f\|_\infty \quad \text{(by } \|\bar{G}'P\bar{G}\mu\|_1 \leq \|\bar{G}'\|_{1\to 1}\|P\|_{1\to 1}\|\bar{G}\|_{1\to 1}\|\mu\|_1 \leq 1\text{)} \\
&\leq (1 - \gamma)\delta_1\|f\|_\infty \tag{D.5}
\end{aligned}$$

Combining equation (D.3) and (D.5) we complete the proof of equation (D.2).

Next we bound $\langle \bar{G}'P\bar{G}\mu, f^2\rangle^{1/2}$ in a more refined manner. By equation (D.1), we have

$$\begin{aligned}
\langle \bar{G}'P\bar{G}\mu, f^2\rangle^{1/2} &= \left(\langle \bar{G}P\bar{G}\mu, f^2\rangle + \frac{1}{1 - \gamma}\langle \bar{G}'\Delta\bar{G}P\bar{G}\mu, f^2\rangle\right)^{1/2} \\
&\leq \langle \bar{G}P\bar{G}\mu, f^2\rangle^{1/2} + (1 - \gamma)^{-1/2}\langle \bar{G}'\Delta\bar{G}P\bar{G}\mu, f^2\rangle^{1/2} \tag{D.6}
\end{aligned}$$

By Lemma D.3 again, we have that

$$\langle \bar{G}'\Delta\bar{G}P\bar{G}, f^2\rangle^2 \leq \chi^2_{\bar{G}P\bar{G}\mu}(P', P)\langle \bar{G}'P\bar{G}P\bar{G}\mu, f^4\rangle \tag{D.7}$$

By Holder inequality and the fact that $\|\bar{G}\|_{1\to 1} = 1$, $\|\bar{G}'\|_{1\to 1} = 1$ and $\|P\|_{1\to 1} = 1$, we have

$$\langle \bar{G}'P\bar{G}P\bar{G}\mu, f^4\rangle \leq \|\bar{G}'P\bar{G}P\bar{G}\mu\|_1\|f^4\|_\infty \leq \|f\|_\infty^4 \tag{D.8}$$

Combining equation (D.6), (D.8) gives

$$(1 - \gamma)^{-1/2}\langle \bar{G}'\Delta\bar{G}P\bar{G}, f^2\rangle^{1/2} \leq ((1 - \gamma)^{-1}\chi^2_{\bar{G}P\bar{G}\mu}(P', P)^{1/2})^{1/2}\|f\|_\infty = \delta_2^{1/2}\|f\|_\infty \tag{D.9}$$

Then, combining equation (D.3), (D.4), (D.9), we have

$$\begin{aligned}
\langle(\bar{G}' - \bar{G})\mu, f\rangle &= (1 - \gamma)^{-1}\chi^2_{\bar{G}\mu}(P', P)^{1/2}\langle \bar{G}'P\bar{G}\mu, f^2\rangle^{1/2} && \text{(by equation (D.3), (D.4))} \\
&= \delta_1\langle \bar{G}'P\bar{G}\mu, f^2\rangle^{1/2} \\
&\leq \delta_1\langle \bar{G}P\bar{G}\mu, f^2\rangle^{1/2} + \delta_1(1 - \gamma)^{-1/2}\langle \bar{G}'\Delta\bar{G}P\bar{G}\mu, f^2\rangle^{1/2} && \text{(by equation (D.6))} \\
&\leq \delta_1\langle \bar{G}P\bar{G}\mu, f^2\rangle^{1/2} + \delta_1\delta_2^{1/2}\|f\|_\infty && \text{(by equation (D.9))}
\end{aligned}$$

$$\square$$

The following Lemma is a stronger extension of Lemma D.4, which can be used to future improve Proposition A.2, and may be of other potential independent interests. We state it for completeness.

**Lemma D.5.** *Let* $\bar{G}, \bar{G}', P', P, f$ *as defined in the beginning of this section. Let* $d_k = (\bar{G}P)^k \bar{G}\mu$ *and* $\delta_k = (1 - \gamma)^{-1}\chi^2_{d_{k-1}}(P', P)^{1/2}$, *then we have that for any* $K$,

$$\left| \langle \bar{G}'\mu, f \rangle - \langle \bar{G}\mu, f \rangle \right| \leq \delta_1 \langle d_1, f^2 \rangle^{-1/2} + \delta_1 \delta_2^{1/2} \langle d_2, f^4 \rangle^{-1/4}$$
$$+ \delta_1 \dots \delta_K^{2^{-K+1}} \langle d_k, f^{2^K} \rangle^{2^{-K}} + \delta_1 \dots \delta_K^{2^{-K+1}} \|f\|_\infty$$

*Proof.* We first use induction to prove that:

$$|\langle (\bar{G}' - \bar{G})\mu, f \rangle| \leq \sum_{k=1}^K \left( \prod_{0 \leq s \leq k-1} \delta_{s+1}^{2^{-s}} \right) \langle d_k, f^{2^k} \rangle^{2^{-k}}$$
$$+ \left( \prod_{0 \leq s \leq K-1} \delta_{s+1}^{2^{-s}} \right) \langle \bar{G}'\Delta(\bar{G}P)^K \bar{G}\mu, f^{2^K} \rangle^{2^{-K}} \quad \text{(D.10)}$$

By the first equation of Lemma D.4, we got the case for $K = 1$. Assuming we have proved the case for $K$, then applying

$$\langle \bar{G}'\Delta(\bar{G}P)^K \bar{G}\mu, f^{2^K} \rangle = \langle \bar{G}\Delta(\bar{G}P)^K \bar{G}\mu, f^{2^K} \rangle + (1 - \gamma)^{-1} \langle \bar{G}\Delta(\bar{G}P)^{K+1} \bar{G}\mu, f^{2^K} \rangle$$
$$\text{(D.11)}$$

$$\leq \langle \bar{G}\Delta(\bar{G}P)^K \bar{G}\mu, f^{2^K} \rangle$$
$$+ (1 - \gamma)^{-1} \chi^2_{d_K}(P', P)^{1/2} \langle \bar{G}'\Delta(\bar{G}P)^{K+1} \bar{G}\mu, f^{2^{K+1}} \rangle^{1/2}$$
$$\leq \langle \bar{G}\Delta(\bar{G}P)^K \bar{G}\mu, f^{2^K} \rangle + \delta_{K+1} \langle \bar{G}'\Delta(\bar{G}P)^{K+1} \bar{G}\mu, f^{2^{K+1}} \rangle^{1/2}$$

By Cauchy-Schwartz inequality, we obtain that

$$\langle \bar{G}'\Delta(\bar{G}P)^K \bar{G}\mu, f^{2^K} \rangle^{2^{-K}} \leq \langle \bar{G}\Delta(\bar{G}P)^K \bar{G}\mu, f^{2^K} \rangle^{2^{-K}} +$$
$$\delta_{K+1} \langle \bar{G}'\Delta(\bar{G}P)^{K+1} \bar{G}\mu, f^{2^{K+1}} \rangle^{2^{-K-1}}$$

Plugging the equation above into equation (D.10), we provide the induction hypothesis for the case with $K + 1$.

Now applying $\langle \bar{G}'\Delta(\bar{G}P)^K \bar{G}\mu, f^{2^K} \rangle^{2^{-K}} \leq \|f\|_\infty$ with equation (D.10) we complete the proof.

$\square$

## E TOOLBOX

**Definition E.1** ($\chi^2$ distance, c.f. Nielsen & Nock (2014); Cover & Thomas (2012)). *The Neyman* $\chi^2$ *distance between two distributions* $p$ *and* $q$ *is defined as*

$$\chi^2(p, q) \triangleq \int \frac{(p(x) - q(x))^2}{q(x)} dx = \int \frac{p(x)^2}{q(x)} dx - 1$$

For notational simplicity, suppose two random variables $X$ and $Y$ has distributions $p_X$ and $p_Y$, we often write $\chi^2(X, Y)$ as a simplification for $\chi^2(p_X, p_Y)$.

**Theorem E.2** (Sason & Verdú (2016)). *The Kullback-Leibler (KL) divergence between two distributions* $p, q$ *is bounded from above by the* $\chi^2$ *distance:*

$$KL(p, q) \leq \chi^2(p, q)$$

*Proof.* Since $\log$ is a concave function, by Jensen inequality we have

$$KL(p, q) = \int p(x) \log \frac{p(x)}{q(x)} dx \leq \log \int p(x) \cdot \frac{p(x)}{q(x)} dx$$

$$= \log(\chi^2(p, q) + 1) \leq \chi^2(p, q)$$

$\square$

**Definition E.3** ($\chi^2$ distance between transitions). Given two transition kernels $P, P'$. For any distribution $\mu$, we define $\chi^2_\mu(P', P)$ as:

$$\chi^2_\mu(P', P) \triangleq \int \mu(x) \chi^2(P'(\cdot|X = x), P(\cdot|X = x)) dx$$

**Theorem E.4.** *Suppose random variables $(X, Y)$ and $(X', Y')$ satisfy that $p_{Y|X} = p_{Y'|X'}$. Then*

$$\chi^2(Y, Y') \leq \chi^2(X, X')$$

*Or equivalently, for any transition kernel $P$ and distribution $\mu, \mu'$, we have*

$$\chi^2(P\mu, P\mu') \leq \chi^2(\mu, \mu')$$

*Proof.* Denote $p_{Y|X}(y \mid x) = p_{Y'|X'}(y \mid x)$ by $p(y \mid x)$, and we rewrite $p_X$ as $p$ and $p_{X'}$ as $p'$. By Cauchy-Schwarz inequality, we have:

$$p_Y(y)^2 = \left( \int p(y|x)p(x)dx \right)^2 \leq \left( \int p(y|x)p'(x)dx \right) \left( \int p(y|x)\frac{p(x)^2}{p'(x)}dx \right)$$

$$= p_{Y'}(y) \left( \int p(y|x)\frac{p(x)^2}{p'(x)}dx \right) \tag{E.1}$$

It follows that

$$\chi^2(Y, Y') = \int \frac{p_Y(y)^2}{p_{Y'}(y)}dy - 1 \leq \int dy \int p(y|x)\frac{p(x)^2}{p'(x)}dx - 1 = \chi^2(X, X')$$

$\square$

**Theorem E.5.** *Let $X, Y, Y'$ are three random variables. Then,*

$$\chi^2(Y, Y') \leq \mathbb{E}\left[\chi^2(Y|X, Y'|X)\right]$$

*We note that the expectation on the right hand side is over the randomness of $X$.[11] As a direct corollary, we have for transition kernel $P'$ and $P$ and distribution $\mu$,*

$$\chi^2(P'\mu, P\mu) \leq \chi^2_\mu(P', P)$$

*Proof.* We denote $p_{Y'|X}(y|x)$ by $p'(y \mid x)$ and $p_{Y|X}(y|x)$ by $p(y|x)$, and let $p(x)$ be a simplification for $p_X(x)$. We have by Cauchy-Schwarz,

$$\frac{p_Y(y)^2}{p_{Y'}(y)} = \frac{\left(\int p(y|x)p(x)dx\right)^2}{\int p'(y \mid x)p(x)dx} \leq \int \frac{p(y|x)^2}{p'(y|x)}p(x)dx$$

It follows that

$$\chi^2(Y, Y') = \int \frac{p_Y(y)^2}{p_{Y'}(y)}dy - 1 \leq \int \frac{p(y|x)^2}{p'(y|x)}p(x)dxdy - 1 = \mathbb{E}\left[\chi^2(Y|X, Y'|X) \mid X\right]$$

$\square$

**Claim E.6.** *Let $\mu$ be a distribution over the state space $\mathcal{S}$. Let $P$ and $P'$ be two transition kernels. $G = \sum_{k=0}^{\infty}(\gamma P)^k = (\text{Id} - \gamma P)^{-1}$ and $G' = \sum_{k=0}^{\infty}(\gamma P')^k = (\text{Id} - \gamma P')^{-1}$. Let $d = (1 - \gamma)G\mu$ and $d' = (1 - \gamma)G'\mu$ be the discounted distribution starting from $\mu$ induced by the transition kernels $G$ and $G'$. Then,*

$$|d - d'|_1 \leq \frac{1}{1 - \gamma}|\Delta d|_1$$

*Moreover, let $\gamma(P' - P) = \Delta$. Then, we have*

$$G' - G = \sum_{k=1}^{\infty}(G\Delta)^k G$$

---

[11]Observe $\chi^2(Y|X, Y'|X)$ deterministically depends on $X$.

*Proof.* With algebraic manipulation, we obtain,

$$G' - G = (\text{Id} - \gamma P')^{-1}((\text{Id} - \gamma P) - (\text{Id} - \gamma P')(\text{Id} - \gamma P)^{-1}$$
$$= G'\Delta G \tag{E.2}$$

It follows that

$$|d - d'|_1 = (1 - \gamma)|G'\Delta G\mu|_1 \le |\Delta G\mu|_1 \qquad (\text{since } (1 - \gamma)|G'|_{1\to 1} \le 1)$$
$$= \frac{1}{1 - \gamma}|\Delta d|_1$$

Replacing $G'$ in the RHS of the equation (E.2) by $G' = G + G'\Delta G$, and doing this recursively gives

$$G' - G = \sum_{k=1}^{\infty}(G\Delta)^k G$$

$\square$

**Corollary E.7.** *Let $\pi$ and $\pi'$ be two policies and let $\rho^\pi$ be defined as in Definition 2.1. Then,*

$$|\rho^\pi - \rho'|_1 \le \frac{\gamma}{1 - \gamma}\mathop{\mathbb{E}}_{S \sim \rho^\pi}\left[KL(\pi(S), \pi'(S))^{1/2} \mid S\right]$$

*Proof.* Let $P$ and $P'$ be the state-state transition matrix under policy $\pi$ and $\pi'$ and $\Delta = \gamma(P' - P)$ By Claim E.6, we have that

$$|\rho^\pi - \rho^{\pi'}|_1 \le \frac{1}{1 - \gamma}|\Delta\rho^\pi|_1 = \frac{\gamma}{1 - \gamma}\mathop{\mathbb{E}}_{S \sim \rho^\pi}\left[|p_{M^\star(S, \pi(S))|S} - p_{M^\star(S, \pi'(S))|S}|_1\right]$$
$$\le \frac{\gamma}{1 - \gamma}\mathop{\mathbb{E}}_{S \sim \rho^\pi}\left[|p_{\pi(S)|S} - p_{\pi'(S)|S}|_1\right]$$
$$\le \frac{\gamma}{1 - \gamma}\mathop{\mathbb{E}}_{S \sim \rho^\pi}\left[KL(\pi(S), \pi'(S))^{1/2} \mid S\right] \quad \text{(by Pinkser's inequality)}$$

$\square$

# F IMPLEMENTATION DETAILS

## F.1 ENVIRONMENT SETUP

We benchmark our algorithm on six tasks based on physics simulator Mujoco (Todorov et al., 2012). We use rllab's implementation (Duan et al., 2016) [12] to interact with Mujoco. All the environments we use have a maximum horizon of 500 steps. We remove all contact information from observation. To compute reward from states, we put the velocity of center of mass into the states.

## F.2 NETWORK ARCHITECTURE AND MODEL LEARNING

We use the same reward function as in rllab, except that all the coefficients $C_{\text{contact}}$ in front of the contact force s are set to $0$ in our case. We refer the readers to (Duan et al., 2016) Supp Material 1.2 for more details. All actions are projected to the action space by clipping. We normalize all observations by $s' = \frac{s - \mu}{\sigma}$ where $\mu, \sigma \in \mathbb{R}^{d_{\text{observation}}}$ are computed from all observations we collect from the real environment. Note that $\mu, \sigma$ may change as we collect new data. Our policy will always produce an action $a$ in $[-1, 1]^{d_{\text{action}}}$ and the action $a'$, which is fed into the environment, is scaled linearly by $a' = \frac{1-a}{2}a_{\text{min}} + \frac{1+a}{2}a_{\text{max}}$, where $a_{\text{min}}, a_{\text{max}}$ are the min or max values allowed at each entry.

---

[12]commit b3a2899 in `https://github.com/rll/rllab/`

Table 1: TRPO Hyperparameters.

| Hyperparameters | Values |
|---|---|
| batch size | 4000 |
| max KL divergence | 0.01 |
| discount $\gamma$ | 0.99 |
| GAE $\lambda$ | 0.95 |
| CG iterations | 10 |
| CG damping | 0.1 |

## F.3  SLBO DETAILS

The dynamical model is represented by a feed-forward neural network with two hidden layers, each of which contains 500 hidden units. The activation function at each layer is ReLU. We use Adam to optimize the loss function with learning rate $10^{-3}$ and $L_2$ regularization $10^{-5}$. The network does not predict the next state directly; instead, it predicts the normalized difference of $s_{t+1} - s_t$. The normalization scheme and statistics are the same as those of observations: We maintain $\mu, \sigma$ from collected data in the real environment and may change them as we collect more, and the normalized difference is $\frac{s_{t+1} - s_t - \sigma}{\mu}$.

The policy network is a feed-forward network with two hidden layers, each of which contains 32 hidden units. The policy network uses $\tanh$ as activation function and outputs a Gaussian distribution $\mathcal{N}(\mu(s), \sigma^2)$ where $\sigma$ a state-independent trainable vector.

During our evaluation, we use $H = 2$ for multi-step model training and the batch size is given by $\frac{256}{H} = 128$, i.e., we enforce the model to see 256 transitions at each batch.

We run our algorithm $n_{\text{outer}} = 100$ iterations. We collect $n_{\text{train}} = 10000$ steps of real samples from the environment at the start of each iteration using current policy with Ornstein-Uhlunbeck noise (with parameter $\theta = 0.15, \sigma = 0.3$) for better exploration. At each iteration, we optimize dynamics model and policy alternatively for $n_{\text{inner}} = 20$ times. At each iteration, we optimize dynamics model for $n_{\text{model}} = 100$ times and optimize policy for $n_{\text{policy}} = 40$ times.

## F.4  BASELINES

**TRPO.**  TRPO hyperparameters are listed at Table 1, which are the same as OpenAI Baselines' implementation. These hyperparameters are fixed for all experiments where TRPO is used, including ours, MB-TRPO and MF-TRPO. We do not tune these hyperparameters. We also normalize observations as our algorithm and OpenAI Baselines do.

We use a neural network as the value function to reduce variance, which has 2 hidden layers of units 64 and uses $\tanh$ as activation functions. We use Generalized Advantage Estimator (GAE) Schulman et al. (2015b) to estimate advantages. Both TRPO used in our algorithm and that in model-free algorithm share the same set of hyperparameters.

**SAC.**  For fair comparison, we do not use a large policy network (2 hidden layers, one of which has 256 hidden units) as the authors suggest, but use exactly the same policy network as ours. All other hyperparameters are kept the same as the authors'. Note that Q network and value network have 256 hidden units at each hidden layers, which is more than TRPO's. We refer the readers to Haarnoja et al. (2018) Appendix D for more details.

**MB-TRPO.**  Model-Based TRPO (MB-TRPO) is similar to our algorithm SLBO but does not optimize model and policy alternatively during one iteration. We do not tune the hyperparameter $n_{\text{model}}$ since any number beyond a certain threshold would bring similar results. For $n_{\text{policy}}$ we try $\{100, 200, 400, 800\}$ on Ant and find $n_{\text{policy}} = 200$ works best in Ant so we use it for all other environments. Note that when Algo 2 uses 800 Adam updates (per outer iteration), it has the same amount of updates (per outer iteration) as in Algo 3. As suggested by Section 6.1, we use 0.005 as the coefficient of entropy bonus for all experiments.

---

**Algorithm 3** Model-Based Trust Region Policy Optimization (MB-TRPO)

---

1: initialize model network parameters $\phi$ and policy network parameters $\theta$
2: initialize dataset $\mathcal{D} \leftarrow \emptyset$
3: **for** $n_{\text{outer}}$ iterations **do**
4:      $\mathcal{D} \leftarrow \mathcal{D} \cup \{$ collect $n_{\text{collect}}$ samples from real environment using $\pi_\theta$ with noises $\}$
5:      **for** $n_{\text{model}}$ iterations **do**
6:          optimize (6.1) over $\phi$ with sampled data from $\mathcal{D}$ by one step of Adam
7:      **for** $n_{\text{policy}}$ iterations **do**
8:          $\mathcal{D}' \leftarrow \{$ collect $n_{\text{trpo}}$ samples using $\widehat{M_\phi}$ as dynamics $\}$
9:          optimize $\pi_\theta$ by running TRPO on $\mathcal{D}'$

---

**SLBO.** We tune multi-step model training parameter $H \in \{1, 2, 4, 8\}$, entropy regularization coefficient $\lambda \in \{0, 0.001, 0.003, 0.005\}$ and $n_{\text{policy}} \in \{10, 20, 40\}$ on Ant and find $H = 2, \lambda = 0.005, n_{\text{policy}} = 40$ work best, then we fix them in all environments, though environment-specific hyperparameters may work better. The other hyperparameters, including $n_{\text{inner}}, n_{\text{model}}$ and network architecture, are never tuned. We observe that at the first several iterations, the policy overfits to the learnt model so a reduction of $n_{\text{policy}}$ at the beginning can further speed up convergence but we omit this for simplicity.

The most important hyperparameters we found are $n_{\text{policy}}$ and the coefficient in front of the entropy regularizer $\lambda$. It seems that once $n_{\text{model}}$ is large enough we don't see any significant changes. We did have a held-out set for model prediction (with the same distribution as the training set) and found out the model doesn't overfit much. As mentioned in F.3, we also found out normalizing the state helped a lot since the raw entries in the state have different magnitudes; if we don't normalize them, the loss will be dominated by the loss of some large entries.

### F.5 ABLATION STUDY

**Multi-step model training.** We compare multi-step model training with single-step model training and the results are shown on Figure 2. Note that $H = 1$ means we use single-step model training. We observe that small $H$ (e.g., 2 or 4) can be beneficial, but larger $H$ (e.g., 8) can hurt. We hypothesize that smaller $H$ can help the model learn the uncertainty in the input and address the error-propagation issue to some extent. Pathak et al. (2018) uses an auto-regressive recurrent model to predict a multi-step loss on a trajectory, which is closely related to ours. However, theirs differs from ours in the sense that they do not use the predicted output $x_{t+1}$ as the input for the prediction of $x_{t+2}$, and so on and so forth.

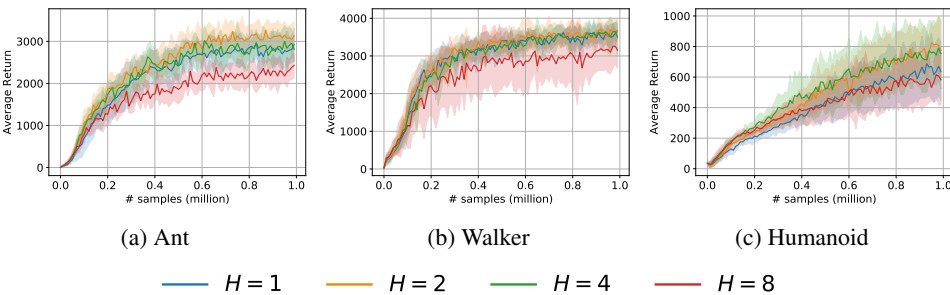

Figure 2: Ablation study on multi-step model training. All the experiments are average over 10 random seeds. The x-axis shows the total amount of real samples from the environment. The y-axis shows the averaged return from execution of our learned policy. The solid line is the mean of the total rewards from each seed. The shaded area is one-standard deviation.

**Entropy regularization.** Figure 3 shows that entropy reguarization can improve both sample efficiency and final performance. More entropy regularization leads to better sample efficiency and higher total rewards. We observe that in the late iterations of training, entropy regularization may hurt the performance thus we stop using entropy regularization in the second half of training.

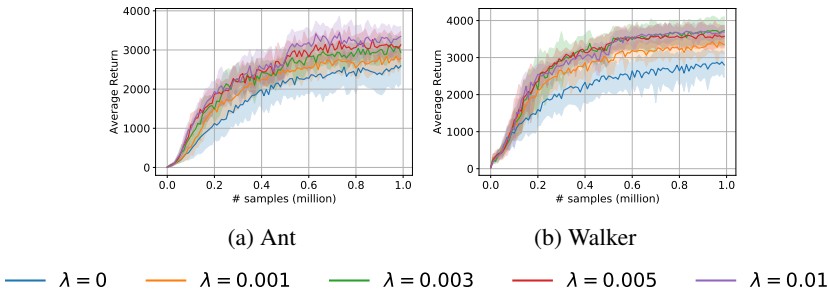

(a) Ant         (b) Walker

Figure 3: Ablation study on entropy regularization. $\lambda$ is the coefficient of entropy regularization in the TRPO's objective. All the experiments are averaged over 10 random seeds. The x-axis shows the total amount of real samples from the environment. The y-axis shows the averaged return from execution of our learned policy. The solid line is the mean of the total rewards from each seed. The shaded area is one-standard deviation.

**SLBO with 4M training steps.** Figure 4 shows that SLBO is superior to SAC and MF-TRPO in Swimmer, Half Cheetah, Walker and Humanoid when 4 million samples or fewer samples are allowed. For Ant environment , although SLBO with less than one million samples reaches the performance of MF-TRPO with 8 million samples, SAC's performance surpasses SLBO after 2 million steps of training. Since model-free TRPO almost stops improving after 8M steps and our algorithms uses TRPO for optimizing the estimated environment, we don't expect SLBO can significantly outperform the reward of TRPO at 8M steps. The result shows that SLBO is also satisfactory in terms of asymptotic convergence (compared to TRPO.) It also indicates a better planner or policy optimizer instead of TRPO might be necessary to further improve the performance.

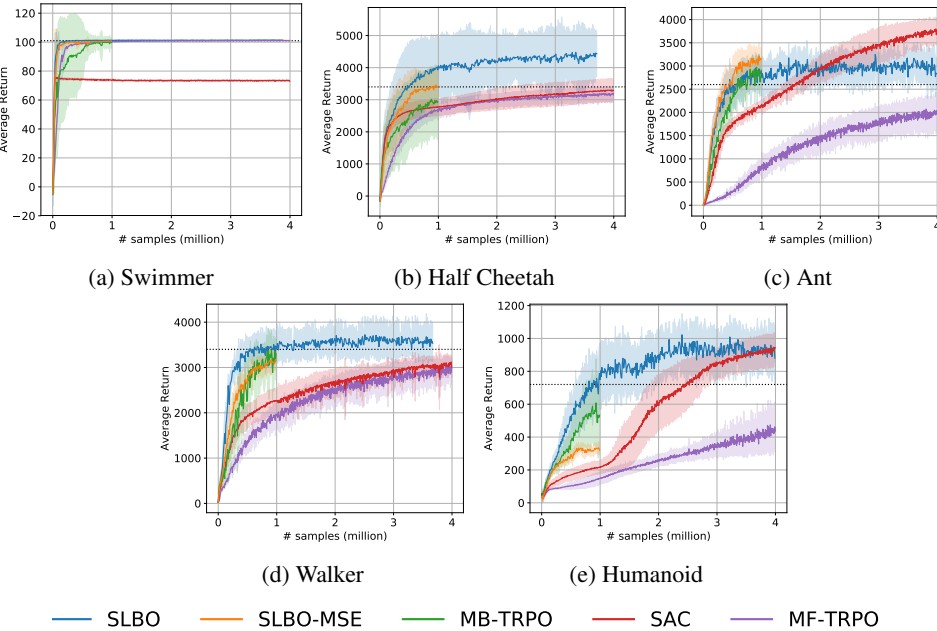

(a) Swimmer       (b) Half Cheetah       (c) Ant

(d) Walker        (e) Humanoid

Figure 4: Comparison among SLBO (ours), SLBO with squared $\ell^2$ model loss (SLBO-MSE), vanilla model-based TRPO (MB-TRPO), model-free TRPO (MF-TRPO), and Soft Actor-Critic (SAC) with more samples than in Figure 1. SLBO, SAC, MF-TRPO are trained with 4 million real samples. We average the results over 10 different random seeds, where the solid lines indicate the mean and shaded areas indicate one standard deviation. The dotted reference lines are the total rewards of MF-TRPO after 8 million steps.

# G    SAMPLE COMPLEXITY BOUNDS

In this section, we extend Theorem 3.1 to a final sample complexity result. For simplicity, let $L_{\pi_{\text{ref}},\delta}^{\pi,M} = V^{\pi,M} - D_{\pi_k,\delta}(M,\pi)$ be the lower bound of $V^{\pi,M^\star}$. We omit the subscript $\delta$ when it's clear from contexts. When $D$ satisfies (R1), we have that,

$$V^{\pi,M^\star} \geq L_{\pi_{\text{ref}},\delta}^{\pi,M} \qquad \forall \pi \text{ s.t. } d(\pi,\pi_{\text{ref}}) \leq \delta \tag{G.1}$$

When $D$ satisfies (R3), we use $\widehat{L}_{\pi_{\text{ref}},\delta}^{\pi,M}$ to denotes its empirical estimates. Namely, we replace the expectation in equation (R3) by empirical samples $\tau^{(1)}, \ldots, \tau^{(n)}$. In other words, we optimize

$$\pi_{k+1}, M_{k+1} = \underset{\pi \in \Pi, \, M \in \mathcal{M}}{\operatorname{argmax}} \widehat{L}_{\pi_{\text{ref}},\delta}^{\pi,M} = V^{\pi,M} - \frac{1}{n}\sum_{i=1}^{n} f(\widehat{M}, \pi, \tau^{(i)}) \tag{G.2}$$

instead of equation (3.3).

Let $p$ be the total number of parameters in the policy and model parameterization. We assume that we have a discrepancy bound $D_{\pi_{\text{ref}}}(\pi, M)$ satisfying (R3) with a function $f$ that is bounded with $[-B_f, B_f]$ and that is $L_f$-Lipschitz in the parameters of $\pi$ and $M$. That is, suppose $\pi$ is parameterized by $\theta$ and $M$ is parameterized by $\phi$, then we require $|f(M_\phi, \phi_\theta, \tau) - f(M_{\phi'}, \phi_{\theta'}, \tau)| \leq L_f(\|\phi - \phi'\|_2^2 + \|\theta - \theta'\|^2)$ for all $\tau, \theta, \theta', \phi, \phi'$. We note that $L_f$ is likely to be exponential in dimension due to the recursive nature of the problem, but our bounds only depends on its logarithm. We also restrict our attention to parameters in an Euclidean ball $\{\theta : \|\theta\|_2 \leq B\}$ and $\{\phi : \|\phi\|_2 \leq B\}$. Our bounds will be logarithmic in $B$.

We need the following definition of approximate local maximum since with sampling error we cannot hope to converge to the exact local maximum.

**Definition G.1.** We say $\pi$ is a $(\delta, \varepsilon)$-local maximum of $V^{\pi,M^\star}$ with respect to the constraint set $\Pi$ and metric $d$, if for any $\pi' \in \Pi$ with $d(\pi, \pi') \leq \delta$, we have $V^{\pi,M^\star} \geq V^{\pi',M^\star} - \varepsilon$.

We show a sample complexity bounds that scales linearly in $p$ and logarithmically in $L_f, B$ and $B_f$.

**Theorem G.2.** *Let $\varepsilon > 0$. In the setting of Theorem 3.1, under the additional assumptions above, suppose we use $n = O(B_f p \log(BL_f/\varepsilon)/\varepsilon^2)$ trajectories to estimate the discrepancy bound in Algorithm 1. Then, for any $t$, if $\pi_t$ is not a $(\delta, \varepsilon)$-local maximum, then the total reward will increase in the next step: with high probability,*

$$V^{\pi_{t+1},M^\star} \geq V^{\pi_t,M^\star} + \varepsilon/2 \tag{G.3}$$

*As a direct consequence, suppose the maximum possible total reward is $B_R$ and the initial total reward is 0, then for some $T = O(B_R/\varepsilon)$, we have that $\pi_T$ is a $(\delta, \varepsilon)$-local maximum of the $V^{\pi,M^\star}$.*

*Proof.* By Hoeffding's inequality, we have for fix $\pi$ and $\widehat{M}$, with probability $1 - n^{O(1)}$ over the randomness of $\tau^{(1)}, \ldots, \tau^{(n)}$,

$$\left| \frac{1}{n}\sum_{i=1}^{n} f(\widehat{M}, \pi, \tau^{(i)}) - \underset{\tau \sim \pi_{\text{ref}}, M^\star}{\mathbb{E}}[f(\widehat{M}, \pi, \tau)] \right| \leq 4\sqrt{\frac{B_f \log n}{n}}. \tag{G.4}$$

In more succinct notations, we have $|\widehat{D}_{\pi_k,\delta}(M, \pi) - D_{\pi_k,\delta}(M, \pi)| \leq 4\sqrt{\frac{B_f \log n}{n}}$, and therefore

$$|\widehat{L}^{\pi,M} - L^{\pi,M}| \leq 4\sqrt{\frac{B_f \log n}{n}}. \tag{G.5}$$

By a standard $\varepsilon$-cover + union bound argument, we can prove the uniform convergence: with high probability (at least $1 - n^{O(1)}$) over the choice of $\tau^{(1)}, \ldots, \tau^{(n)}$, for all policy and model, for all policy $\pi$ and dynamics $M$,

$$|\widehat{L}^{\pi,M} - L^{\pi,M}| \leq 4\sqrt{\frac{B_f p \log(nBL_f)}{n}} = \varepsilon/4. \tag{G.6}$$

Suppose at iteration $t$, we are at policy $\pi_t$ which is not a $(\delta, \varepsilon)$-local maximum of $V^{\pi, M^\star}$. Then, there exists $\pi'$ such that $d(\pi', \pi_t) \leq \delta$ and

$$V^{\pi', M^\star} \geq V^{\pi_t, M^\star} + \varepsilon. \tag{G.7}$$

Then, we have that

$$
\begin{aligned}
V^{\pi_{t+1}, M^\star} &\geq L_{\pi_t}^{\pi_{t+1}, M_{t+1}} && \text{(by equation (G.1))} \\
&\geq \widehat{L}_{\pi_t}^{\pi_{t+1}, M_{t+1}} - \varepsilon/4 && \text{(by uniform convergence, equation (G.6))} \\
&\geq \widehat{L}_{\pi_t}^{\pi', M^\star} - \varepsilon/4 && \text{(by the definition of } \pi_{t+1}, M_{t+1}) \\
&\geq L_{\pi_t}^{\pi', M^\star} - \varepsilon/2 && \text{(by uniform convergence, equation (G.6))} \\
&= V^{\pi', M^\star} - \varepsilon/2 && \text{(by equation (R2))} \\
&= V^{\pi_t, M^\star} + \varepsilon/2 && \text{(by equation (G.7))}
\end{aligned}
$$

Note that the total reward can only improve by $\varepsilon/2$ for at most $O(B_R/\varepsilon)$ steps. Therefore, in the first $O(B_R/\varepsilon)$ iterations, we must have hit a solution that is a $(\delta, \varepsilon)$-local maximum. This completes the proof.

$\square$

