# OpenReview forum: "Algorithmic Framework for Model-based Deep Reinforcement Learning with Theoretical Guarantees"
_ICLR.cc/2019/Conference_

### Official Review · AnonReviewer3 · 2018-11-02
**A novel framework for deep RL but needs more specific examples**

**Rating:** 6
**Confidence:** 2

**Review:**

This paper proposed a new class of meta-algorithm for reinforcement learning and proved the monotone improvement for a local maximum of the expected reward, which could be used in deep RL setting. The framework seems to be quite general but does not include any specific example, like what non-linear dynamical model in detail could be included and will this framework cover the classical MDP setting? In theory, the dynamical model needs to satisfy L-Lipschitz. So which dynamical model in reality could satisfy this assumption? It seems that the focus of this paper is theoretical side. But the only guarantee is the non-decreasing value function of the policy. In RL, people may be more care about the regret or sample complexity. Previous model-based work with simpler model already can have such strong guarantees, such as linear dynamic (Y. Abbasi-Yadkori and Cs. Szepesvari (2011)), MDP (Agrawal and Jia (2017)). What kind of new insights will this framework give when the model reduces to simpler one (linear model)?

In practical implementation, the authors designed a Stochastic Lower Bound Optimization. Is there any convergence rate guarantee for this stochastic optimization? And also neural network is used for deep RL. So there is also no guarantee for the actual algorithm which is used?

Minor:

1. In (3.2), what norm is considered here?
2. In page 4, the authors mentioned their algorithm can be viewed as an extension of the optimism-in-face-of-uncertainty principle to non-linear parameterized setting. This is a little bit confused. How this algorithm can be viewed as OFU principle? How does it recover the result in linear setting (Y. Abbasi-Yadkori and Cs. Szepesvari (2011))?
3. The organization could be more informative. For example, Section 1 has 13 paragraphs but without any subsection.

Y. Abbasi-Yadkori and Cs. Szepesvari, Regret Bounds for the Adaptive Control of Linear Quadratic Systems, COLT, 2011.
Shipra Agrawal and Randy Jia. Optimistic posterior sampling for reinforcement learning: worst-case regret bounds. NIPS, 2017

---

> ### Author Response · Authors · 2018-11-09
> **Response**
>
> We thank the reviewer for the insightful review and positive comments on the theoretical framework. We address the reviewer’s comments/questions below:
>
> --- “The framework seems to be quite general but does not include any specific example, like what non-linear dynamical model in detail could be included and will this framework cover the classical MDP setting”, “Previous model-based work with simpler model already can have such strong guarantees, such as linear dynamic (Y. Abbasi-Yadkori and Cs. Szepesvari (2011)), MDP (Agrawal and Jia (2017)). What kind of new insights will this framework give when the model reduces to simpler one (linear model)?”
>
> Indeed, our framework can capture all parameterized models (including linear model or even tabular MDP); however, our focus is on non-linear models. The distinction to the previous papers is that we are the first framework that can show the monotone improvement and handle the uncertainty quantification (via a discrepancy bound) *for non-linear models*. As far as we understand, the existing papers’ techniques are difficult to extend to non-linear models. Our approach, restricted to linear models or classical MDP, would give some sensible results but wouldn’t be as strong as the existing ones, and would probably not provide much more insights. However, the strength of the paper is that it works for non-linear models and the key insight is that we don’t need explicit uncertainty quantification of the parameters in the traditional sense (instead, a discrepancy bound would suffice.)
>
> --- “in RL, people may be more care about the regret or sample complexity. ”
>
> We can actually prove a polynomial (in dimension) sample complexity bound for Algorithm 1 with very standard concentration inequality. We can prove uniform convergence results with standard machinery for the estimation of the discrepancy bounds via samples when the bound satisfies R3. Then we can show that the algorithm converges to an approximate local maximum with an error that depends on the estimation error of the discrepancy bound. Such a polynomial complexity bound will not be comparable to Y. Abbasi-Yadkori and Cs. Szepesvari (2011) when restricted to linear models, but they can work generically for non-linear models (under the assumption of Theorem 3.1.) This result is not written in the paper because we thought it’s relatively standard, but we would be more than happy to add it in the revision very soon.
>
> --- “1. In (3.2), what norm is considered here?”
>
> Equation (3.2) is a demonstration of a potential type of results we could hope for. In Section 4, we show that if the value function is L-Lipschitz with some norm, then (3.2) would be true with the same norm. In the experiments, we use the L2 norm.
>
> --- “2. In page 4, the authors mentioned their algorithm can be viewed as an extension of the optimism-in-face-of-uncertainty principle to non-linear parameterized setting. This is a little bit confused. How this algorithm can be viewed as OFU principle? How does it recover the result in linear setting (Y. Abbasi-Yadkori and Cs. Szepesvari (2011))?”
>
> The relationship to OFU is in the very conceptual sense that we optimize the model and the policy together in an optimistic fashion as in OFU. However, the way to quantify the uncertainty is through the discrepancy bound but not the confidence interval as in typical OFU approaches. (But many OFU based papers, such as Jaksch et al’10, implicitly uses some sort of discrepancy bound that is similar to ours in nature in their proof techniques.)
>
> --- “- Is there any convergence rate guarantee for this stochastic optimization? ” “And also neural network is used for deep RL. So there is also no guarantee for the actual algorithm which is used?”
>
> The concrete implementation of the algorithm doesn’t have a convergence rate guarantee yet. We don’t expect it to work for all environments, but under some assumptions of the environments, we may be able to show convergence. This is left as future work.
>
> We also thank the reviewer for the suggestions to add sub-sections in Section 1 and will revise in the next revision. We will also cite the two relevant papers mentioned by the reviewers in the revision.

---

> ### Author Response · Authors · 2018-11-19
> **Response 2: Sample Complexity**
>
> We’ve added a paragraph below Theorem 3.1 and Appendix G, which contains a finite sample complexity results. We can obtain an approximate local maximum in $O(1/\epsilon)$ iterations with sample complexity (in the number of trajectories) that is linear in the number of parameters and accuracy $\epsilon$ and is logarithmic in certain smoothness parameters.
>
> We note that the bound doesn’t directly apply to LQR because we require the reward is bounded (which is not true in LQR because the states can blow up.) If the reward (at a single step) is bounded --- which is a reasonable assumption for many practical applications ---  then our sample complexity can be better (or at least comparable) to Abbasi-Yadkori, Szepesvari in dimension (it’s not clear how AS’s bound depends on the dimension --- it’s likely exponential in dimension.) We also note that AS applies to the adaptive setting which is stronger than the episodic setting that we work with. Finally, we’d like to mention again that our paper’s primary goal is to deal with non-linear setting without explicit uncertainty quantification. In this sense, our result is much stronger than AS because our result applies to any non-linear models with bounded rewards.

---

### Official Review · AnonReviewer2 · 2018-11-04
**Very nice theoretical framework for model-based RL and also an interesting algorithm with promising results is presented. However, there is a large mismatch between the assumptions of the theory and the assumptions made for the algorithm such that it is unclear how much theory can still be used to characterize this algorithm.**

**Rating:** 6
**Confidence:** 4

**Review:**

The paper presents monotonic improvement bounds for model-based reinforcement learning algorithms. Based on these bounds, a new model-based RL algorithm is presented that performs well on standard benchmarks for deep RL.

The paper is well written and the bounds are very interesting. The algorithm is also interesting and seems to perform well. However, there is a slight disappointment after reading the paper because the resulting algorithm is actually quite far away from the assumptions made for deriving the bounds. The 2 innovations of the algorithm are:
- Model and policy are optimized iteratively in an inner policy improvement loop. As far as I see it, this is independent of the presented theory.
- The L2 norm is used to learn the model instead of the squared L2 norm. This is inspired by the theory.

More comments below:
- I was confused by section 4.2. Could you please explain why the transformation is needed and how it is used? As I understand, this is not used at all in the algorithm any more? So what is the advantage of this derivation in comparison to Eq 4.6?
- Please explain in more detail what the effects are from relaxing the assumptions for the algorithm? I assume none of the monotonic improvement results can be transferred to the algorithm?
- Could you elaborate why the algorithm was not implemented as suggested by Section 4? Is the problem that the algorithm did not perform well or that the discrepency measure is hard to compute?
- For the presented algorithm, the discrepency does not depend on the policy any more. I did not understand why the iterative optimization should be useful in this case.
- The theory suggests that we have to do a combined optimization of the lower bound. However, effectively, the algorithm optimizes the policy over V and the policy over the L2 multi-step prediction loss. The difference to a standard model-based RL algorithm is minor and the many advantages of the nice theoretical framework are lost.
- The only difference between Algo 3 and Algo 2 seems to be the additional for loop. As I said, its not clear to me why this should be useful as the optimization problems are independent of each other (except for the trajectories, but the model does not depend on the policy). Did you try Algo 3 with the same amount of Adam updates as Algo 2 (could be that I missed that).

---

> ### Author Response · Authors · 2018-11-09
> **Response - Part 1/2**
>
> We thank the reviewer for the insightful review and positive comments on the theoretical framework. We address the reviewer’s comments/questions below:
>
> --- It seems that the reviewer thinks our empirical implementation is different from what the theory suggests: “ the resulting algorithm is actually quite far away from the assumptions made for deriving the bounds”.
>
> We would like first to mention/clarify that our proposed algorithm (Algorithm 1) is a meta-algorithm/framework for model-based RL. Our main goal is to develop some framework to mathematically reason about non-linear MB RL (such as how to design the model loss function.) The meta-algorithm is designed to have provable monotone convergence, even for the worst-case environments. However, in the empirical implementation, since MuJoCo tasks have nice properties (e.g., the value functions tend to be Lipschitz in states), many components of the meta-algorithm are not necessary, and thus we only need a simplification of the meta-algorithm with a simple discrepancy bound in Section 4.1.
>
> We tried hard to find the simplest instantiation of our meta-algorithm for MuJoCo tasks, instead of using an artificially complicated algorithm. That doesn’t necessarily mean that other instantiations wouldn’t work. (In fact, as mentioned below, some others are promising, though not entirely successful yet. Our current implementation also mostly just serves as a proof-of-concept demonstration that some instantiations of the framework are possible and helpful.)
> The theoretical results in MBRL are very sparse. To some extent, we hope that our work can spark future works that either instantiate our meta-algorithm with strong and clever modifications or that improve our meta-algorithm with stronger guarantees.
>
> Moreover, we would like to argue that the two new empirical ingredients pointed out by the reviewer are both inspired by the theory, in our opinion and our research process. First, the technique of optimizing the policy and model iteratively in an inner policy improvement loop may sound unrelated to the theory, but actually, it was very much inspired by it: our theory suggests that we should jointly optimize the model and the policy to maximize the lower bound for the real reward by SGD, and this would have perfectly justified the iterative optimization of the policy and the model in an inner loop. Later in the experiments, we found that stopping the gradient from one occurrence of the model parameter would not hurt the performance and would speed up the code. Doing so would a priori implies that alternating updates of the model and the policy in an inner loop are less useful, but in fact, the stochasticity introduced by the SGD on model loss is still powerful to reduce overfitting, in a way similar to that SGD regularizes the ordinary supervised training. (Please see paragraph before section 6.2, or the response below to the last two questions,  for slightly more detailed discussions.) Therefore, we view this technique as inspired crucially by the theory, though disguised by the simplification of our algorithm.
>
> As the reviewer agreed, the use of L2 norm (instead of MSE) is inspired and justified by the theory and it also contributes significantly to the empirical improvements.
>
> ---  “I was confused by section 4.2. Could you please explain why the transformation is needed and how it is used?”
>
> The transformation is only to demonstrate that the norm-based model loss is not invariant to a potential hidden transformation of the state space, whereas the discrepancy bound proposed in Section 4.2 is. This is a feature of the algorithm:  if somehow the algorithm is presented with states in different representation space, it will still work the same, whereas the norm-based model loss will behave differently. If one is not concerned with the representation of the states, this section indeed only provides the formal error bound of the discrepancy bound in equation 4.6.

---

> ### Author Response · Authors · 2018-11-09
> **Response - Part 2/2**
>
>
> --- “Please explain in more detail what the effects are from relaxing the assumptions for the algorithm? I assume none of the monotonic improvement results can be transferred to the algorithm?”
>
> We will still have monotone improvements if the algorithm uses any of the discrepancy bounds in Section 4. The monotonicity won’t hold in the worst case, if the model is not optimized in an optimistic fashion. The worst-case scenario would be that the lower bound is quite loose for some dynamical models, and accurate for others. In this case, we would really have to be optimistic about the lower bound and choose the best models and policy. However, such situations are unlikely to occur in MuJoCo tasks since the looseness of the lower bound seems to be comparable in a neighborhood of the current model. This may be the reason why we can simplify the algorithm.
>
>
> ---  “Could you elaborate why the algorithm was not implemented as suggested by Section 4? Is the problem that the algorithm did not perform well or that the discrepancy measure is hard to compute?”
>
> We implemented the discrepancy bound in Section 4.1 as reported in the experiments. The discrepancy measure of Section 4.2 involves the value function which requires another neural net approximators (and thus the resulting algorithm would update the model, the value, and the policy iteratively.) We have implemented this algorithm and it works fine but not as well as the reported simpler version. This may be because either the MuJoCo environments satisfy the assumption in Section 4.1 well, so that L2 norm model loss already performs great, or we have not pinned down the best ways to combine the updates of the model, value, and policy iteratively.
>
> --- “For the presented algorithm, the discrepancy does not depend on the policy any more. I did not understand why the iterative optimization should be useful in this case.”
>
> As briefly mentioned in one of the previous paragraphs, the key benefit of iterative optimization is from the stochasticity in the model when we optimize the imaginary value V^{\pi, M} over the policy. In other words, if we were to optimize M until convergence and then optimize pi, we may optimize the lower bound better, but the algorithm doesn’t use samples in a fully stochastic way. The stochasticity dramatically reduces the overfitting (of the policy to the estimated dynamical model) in a way similar to that SGD regularizes ordinary supervised training. To some extent, since the policy optimization involves stochastic iterates from the updates of the model learning loss, the learned policy has to be robust to a family of stochastic models instead of a single one.
>
> --- “The only difference between Algo 3 and Algo 2 seems to be the additional for loop. ….. Did you try Algo 3 with the same amount of Adam updates as Algo 2 (could be that I missed that). ” “The difference to a standard model-based RL algorithm is minor and the many advantages of the nice theoretical framework are lost”
>
> Indeed we did try Algo 3 with the same amount of Adam updates as in Algo 2, and it performs worse than the current setting.  In fact, we used the optimal number of Adam updates in Algo 3.  Concretely,  we test the performance of Algo 3 with different hyper-parameters, including the number of Adam updates. Our experiments show that 200 (among 100, 200, 400, 800) is the optimal number of Adam updates in Ant and we use it for all other environments. Note that when Algo 3 uses 800 Adam updates (per outer iteration), it has the same amount of updates (per outer iteration) as in Algo 2.
>
> Therefore, the differences of ours from the standard MBRL algorithms, though look simple, are empirically important for the significant improvements of the performance. As we try to argue above, these differences were indeed inspired by the theory.

---

### Official Review · AnonReviewer4 · 2018-11-12
**Interesting theoretical framework for model-based RL and convincing results. It can be improved by adding some clarifications and connections with other RL methods.**

**Rating:** 7
**Confidence:** 4

**Review:**

The paper proposed a framework to design model-based RL algorithms. The framework is based on OFU and within this framework the authors develop an algorithm (a variant of SLBO) achieving SOTA performance on MuJoCo tasks.

The paper is very well written and the topic is important for the RL community. The authors do a good job at covering related works, the bounds are very interesting and the results quite convincing.

Questions/comments to the authors:
1) In footnote 3 you state that "[...] we only need to approximate the dynamical model accurately on the trajectories of the optimal policy". Why only of the optimal policy? Don't you also need an accurate dynamic model for the current policy to perform a good policy improvement step?
2) A major challenge in RL is that the state distribution \rho^\pi changes with \pi and it is usually very hard to estimate. Therefore, many algorithms assume it does not change if the policy is subject to small changes (examples are PPO and TRPO). In Eq 4.3 it seems that you also do something similar, fixing \rho^\pi and constraining the KL of \pi (and not of the joint distribution p(s,a)). Am I correct? Can you elaborate it a bit more, building a connection with other RL methods?
3) In Eq. 6.1 and 6.2 you minimize the H-loss, defined as the prediction error of your model. Recently, Pathak et al. used the same loss function in many papers (such as Curiosity-driven Exploration by Self-supervised Prediction) and your Eq. 6.2 looks like theirs. The practical implementation of your algorithm looks very similar to theirs too. Can you comment on that?
4) If I understood it correctly, your V-function directly depends on your model, i.e., you have V(M(s)) and you learn the model M parameters to maximize V. This means that you want to learn the model that, together with the policy, maximizes V. Am I correct? Can you comment a bit more on that? Did you try to optimize them (V and M) separately, i.e., to add a third parameter to learn (the V-function parameters)?
5) How does you algorithm deal with environmental noise? The tasks used for the evaluation are all deterministic and I believe that this heavily simplifies the model learning. It would be interesting an evaluation on a simple problem (for example the swing-up pendulum) in the presence of noise on the observations and/or the transition function.
6) I appreciate that you provide many details about the implementation in the appendix. Can you comment a bit more? Which are the most important hyperparameters? The number of policy optimization n_policy or of model optimization n_model? You mention that you observed policy overfitting at the first iterations. Did you also experience model overfitting? Did normalizing the state help a lot?

---

> ### Author Response · Authors · 2018-11-16
> **Response - Part 1/2**
>
> We thank the reviewer for the insightful and positive comments. We address the questions below:
>
> 1) “In footnote 3 you state that "[We note that such an assumption, though restricted, may not be very far from reality: optimistically speaking], we only need to approximate the dynamical model accurately on the trajectories of the optimal policy". Why only of the optimal policy? Don't you also need an accurate dynamic model for the current policy to perform a good policy improvement step?”
>
> In the most optimistic scenario, one only needs a not-so-accurate model around the trajectories of a non-optimal policy to make *some reasonable* progress. We note that it’s likely preferable to make decent progress with a non-perfect model compared to making optimal progress with a perfect model, because learning perfect models would require much more samples.
>
> 2) A major challenge in RL is that the state distribution \rho^\pi changes with \pi and it is usually very hard to estimate. Therefore, many algorithms assume it does not change if the policy is subject to small changes (examples are PPO and TRPO). In Eq 4.3 it seems that you also do something similar, fixing \rho^\pi and constraining the KL of \pi (and not of the joint distribution p(s,a)). Am I correct? Can you elaborate it a bit more, building a connection with other RL methods?
>
> You are correct that we constrain the changes of  \rho^\pi. We compare with PPO and TRPO from this perspective in the remark 4.5. We summarize the key point here (please see Remark 4.5 for a longer and more technical discussion): the main advantage of MB approach to TRPO is that our constraint on the changes of \rho^\pi can be more relaxed than that in TRPO. Or in other words, the sensitivity of the reward approximation to the change of \rho^\pi is smaller in our algorithm than in TRPO. This is mostly because, in MB algorithms, the approximation error of the total reward by the imaginary total reward decreases as the model error decreases (even with a fixed change of \rho^\pi), whereas, in model-free algorithms, the approximation error of the total reward by the local linear approximation only depends on the change of the \rho^\pi. Intuitively, we build a better local approximation of the reward using the models than the linear approximation in TRPO.
>
> 3) In Eq. 6.1 and 6.2 you minimize the H-loss, defined as the prediction error of your model. Recently, Pathak et al. used the same loss function in many papers (such as Curiosity-driven Exploration by Self-supervised Prediction) and your Eq. 6.2 looks like theirs. The practical implementation of your algorithm looks very similar to theirs too. Can you comment on that?
>
> This H-loss is not a contribution of ours (e.g., as mentioned in our paper, it has been used in Nagabandi et al’‎2017 for evaluation.) Our implementation differs from “Curiosity-driven Exploration by Self-supervised Prediction” in the sense that we consider the prediction after multiple steps while theirs only considers the prediction of next state (thus one-step prediction).  The Zero-shot visual imitation learning paper by Pathak et al uses an auto-regressive recurrent model to predict a multi-step loss on a trajectory, which is closely related to ours. However, theirs differ from ours in the sense that they do not use the predicted output x_{t+1} as the input for the prediction of x_{t+2}, and so on and so forth. Thanks for pointing out the reference! We include this work in our references and discuss more in our next revision.

---

> ### Author Response · Authors · 2018-11-16
> **Response - Part 2/2**
>
>
> 4) If I understood it correctly, your V-function directly depends on your model, i.e., you have V(M(s)) and you learn the model M parameters to maximize V. This means that you want to learn the model that, together with the policy, maximizes V. Am I correct? Can you comment a bit more on that? Did you try to optimize them (V and M) separately, i.e., to add a third parameter to learn (the V-function parameters)?
>
> Yes, V-function directly depends on your model and we learn the M parameters and \pi parameters to maximize V. In other words, in our current implementation, we don’t have a parameterized approximator for V, and V is computed by querying the model. It’s a fascinating idea of using a third function approximator for V and learn that as well. This is left as future work though.
>
>
> 5) How does you algorithm deal with environmental noise? The tasks used for the evaluation are all deterministic and I believe that this heavily simplifies the model learning. It would be interesting an evaluation on a simple problem (for example the swing-up pendulum) in the presence of noise on the observations and/or the transition function.
>
> The MuJoCo locomotion environments are deterministic yet very challenging. The dynamics of such environments are very complex (e.g. the humanoid dynamics) thus this demonstrates the effectiveness of our method. Many of reinforcement learning algorithms are using these locomotion environments as testbeds. Our meta-algorithms also applies to stochastic environments. We will try to apply the algorithm to a stochastic environment empirically, and hopefully, we can add this to the revision soon.
>
> 6) I appreciate that you provide many details about the implementation in the appendix. Can you comment a bit more? Which are the most important hyperparameters? The number of policy optimization n_policy or of model optimization n_model? You mention that you observed policy overfitting at the first iterations. Did you also experience model overfitting? Did normalizing the state help a lot?
>
> The most important hyperparameters we found are n_policy and the coefficient in front of the entropy regularizer. It seems that once n_model is large enough we don’t see any significant changes. We did have a held-out set for model prediction (with the same distribution as the training set) and found out the model doesn’t overfit much. Normalizing the state helped a lot since the raw entries in the state have different magnitudes ---  if we don’t normalize them, the loss will be dominated by the loss of some large entries.

---

> > ### Comment · AnonReviewer4 · 2018-11-25
> > **Response to rebuttal**
> >
> > Thank you for answering my questions. I have adjusted my score accordingly.

---

### Author Response · Authors · 2018-11-19
**Revision**

As promised in the responses to the reviewers, we have updated the paper with the following changes:

--- We’ve added the citations mentioned by the reviewers, and incorporated most of the clarifications in the responses in the paper. (E.g., in Appendix F.4, we discussed more on the most important hyperparameters.)

--- We’ve added a paragraph below Theorem 3.1 and Appendix G, which contains a finite sample complexity results. We can obtain an approximate local maximum in $O(1/\epsilon)$ iterations with sample complexity (in the number of trajectories) that is linear in the number of parameters and accuracy $\epsilon$ and is logarithmic in certain smoothness parameters.

---

### Meta-Review · Area_Chair1 · 2018-12-16
**Good paper**

**Confidence:** 2
**Recommendation:** Accept (Poster)

**Metareview:**

This paper proposes model-based reinforcement learning algorithms that have theoretical guarantees. These methods are shown to good results on Mujuco benchmark tasks. All of the reviewers have given a reasonable score to the paper, and the paper can be accepted.